# The impact of the Syrian conflict on population well-being

Felix Cheung [1,2✉], Amanda Kube [3], Louis Tay [4], Edward Diener[5], Joshua J. Jackson [6], Richard E. Lucas [7], Michael Y. Ni [1,8,9,10] & Gabriel M. Leung [1,10]

The United Nations described the Syrian conflict as the worst man-made disaster since World War II. We adopted a global perspective in examining the impact of the Syrian conflict on Syrians' physical, mental, and social well-being using the Gallup World Poll. Face-to-face interview data of 11,452 Syrian participants from 2008 to 2015 show that Syrians' physical (e.g., access to shelter), mental (e.g., life satisfaction), and social (e.g., social support) well-being decline substantially. Syrians who reported being exposed to the conflict are similarly affected compared to those without direct exposure, suggesting country-wide spillover effects. Global data covering 1.7 million participants across 163 countries from 2006 to 2016 show during the conflict, Syria's precipitous decline in well-being is unparalleled in the world, even when compared to countries similarly experiencing war, protests, and disasters. Our findings reinforce the vital importance of an accelerated peace process to restore well-being in Syria.

[1] School of Public Health, LKS Faculty of Medicine, University of Hong Kong, 1/F Patrick Manson Building (North Wing), 7 Sassoon Road, Pokfulam, Hong Kong, China. [2] Department of Psychology, University of Toronto, Sidney Smith Hall, 100 St. George Street, Toronto, NO M5S 3G3, Canada. [3] Division of Computational and Data Sciences, Washington University in St. Louis, 1 Brookings Dr., St. Louis, MO 63130, USA. [4] Department of Psychological Sciences, Purdue University, 703 Third Street, West Lafayette, IN 47907-2081, USA. [5] Department of Psychology, University of Utah, University of Virginia, P. O. Box 400400, Charlottesville, VA 22904-4400, USA. [6] Department of Psychological and Brain Sciences, Washington University in St. Louis, 1 Brookings Dr., St. Louis, MO 63130, USA. [7] Department of Psychology, Michigan State University, 316 Physics Rd., East Lansing, MI 48824, USA. [8] The State Key Laboratory of Brain and Cognitive Sciences, The University of Hong Kong, Hong Kong Special Administrative Region, Pokfulam, Hong Kong, China. [9] Healthy High Density Cities Lab, HKUrbanLab, The University of Hong Kong, SAR, Hong Kong, People's Republic of China. [10]These authors contributed equally: Michael Y. Ni, Gabriel M. Leung. ✉email: f.cheung@utoronto.ca

The on-going Syrian conflict began in 2011 as a civil uprising rooted in the Arab Spring and escalated to an armed conflict with targeted strikes on healthcare facilities and personnel, terrorism, and chemical attacks. During the course of the conflict, at least 400,000 Syrians have lost their lives, and 5.6 million refugees out of a pre-war population of 22 million were displaced from Syria[1-3]. The United Nations initially described the Syrian conflict as the "worst man-made disaster the world has seen since World War II", but has since released a blank statement, signalling that words can no longer describe the crisis in Syria[4,5].

The Syrian conflict has widespread consequences on civilian health and well-being beyond mortality and displacement. Due to widespread destruction of health care facilities, progress in infant mortality has reversed, infectious diseases have been on the rise, and patients with chronic diseases have lost access to treatment[6-8]. The damage to residential buildings and food distribution systems disrupt the basic needs for shelter and food[9]. Exposure to war-related brutality is a salient risk factor for debilitating psychological conditions, with long-term impact that usually persists beyond the course of the war[10]. Armed conflicts also destroy families and communities and force civilians into displacement, thus severely compromising social capital[9,11]. Based on prior findings from major disasters, younger age, being female, exposure to the conflict, and lack of social support are additional risk factors for well-being decrement.[9-12] However, objective assessment of population well-being is obstructed by the unprecedented "weaponisation of health care" (i.e., the denial of access to health care as a strategy of war)[13,14]. Therefore, Syrians' self-reported well-being could be particularly relevant to measuring an important aspect of the impact of the conflict. Based on the World Health Organization's holistic definition of health, we sought to assess the changes in self-reported physical, mental, and social well-being and to characterise well-being by sex, age, social support, and exposure to the conflict[15].

Although a large corpus of research has focused on individual major population events[13,16-19], comparison across different events is rarely conducted. Notably, in mental well-being research, studies have found little to no long-term decrease following the 2007-2008 global financial crisis, 2011 Fukushima disaster, or 2015 Paris terrorist attack[20-23]. These findings give evidentiary support to an influential hypothesis in the psychology literature that life circumstances may play a limited role in changing long-term well-being[24]. However, these findings do not appear to be consistent with empirical research on disaster and health[10]. To clarify such apparent divergence, we adopted a global perspective to evaluate the changes in well-being in Syria relative to the rest of the world, including comparisons with other countries' experiences with armed conflict, social unrest, and natural disasters

In the current study, we retrieve face-to-face interview data from random stratified samples of Syrian participants ($n = 11,452$) in the 2006–2016 Gallup World Poll to examine the well-being sequelae of the Syrian conflict[25,26]. Participants' physical (e.g., access to healthcare, food, and shelter), mental (e.g., life satisfaction), and social (e.g., social support) well-being are assessed using 13 indicators. Here we report substantial drops in well-being in all but 2 outcomes based on multilevel models. The declines are population-wide, affecting men and women, young and old, those reported direct or no exposure, and those with or without social support. We then place the impact of the Syrian conflict into global perspective using cross-national data ($n = 1,722,558$) from the Gallup World Poll, a dataset that is representative of 99% of the world's adult population[25,26]. This research design allows us to conduct a comprehensive and global examination of the consequence of the Syrian conflict. The well-

being reduction in 11 outcomes observed in Syria exceeded all other countries surveyed in the Gallup World Poll, even when compared with other countries in turmoil.

## Results

**Serial cross-sectional individual-level analyses.** Table 1 presents the questionnaire items assessing the key variables in the current analyses. Descriptive statistics of the Syrian sample are presented in Table 2, and Fig. 1 shows the geographical distribution of the participants. The longitudinal trends of physical, mental, and social well-being in Syria are shown in Fig. 2.

Overall, 11 out of 13 well-being measures (with the exception of health problems and physical pain) worsened over time (Fig. 2, Table 2, and Supplementary Table 1). From 2008 to 2015, participants were 3.57 (95% CI 3.13 to 4.17) times more likely to report being dissatisfied with healthcare. Difficulties with access to food (OR = 6.74, 95% CI 5.67 to 8.02) and shelter (OR = 3.45, 95% CI 2.94 to 4.05) also increased very substantially from 2008 to 2015. The absolute prevalence of negative emotions increased by 41.4% (95% CI 39.3% to 43.5%). Life satisfaction (range: 1–10) was halved from 2008 (5.15, 95% CI 5.06 to 5.24) to 2015 (2.55, 95% CI 2.32 to 2.78). During the conflict, Syrians were 5-times less likely (OR = 0.20, 95% CI 0.17 to 0.23) to report having someone to count on compared to before the conflict.

**Demographic and geographical differences in well-being.** We explored the sex, age, and geographical differences in these well-being trends (full regression results available in Supplementary Tables 2–4). Overall, sex and age differences in well-being trends were small (for plots, see Supplementary Figs. 1 and 2). In terms of geographical differences, some measures showed substantial variations across governorates (Fig. 1). From 2008 to 2015, participants from Rural Damascus (where the death toll was highest based on estimates from 2011 to 2014)[27] were 15 times (95% CI 13.9 to 17.5) less likely to report satisfaction with access to healthcare, compared to 3.6 times (95% CI 3.1 to 4.2) for the overall country. Aleppo, where the battle accounted for about one-tenth of the death toll and caused most damage to hospitals[13,28], showed the sharpest decline in hope from 2008 to 2015, and the extent of the estimated decline (−2.22 unit, 95% CI −2.29 to −2.16) was about 10 times that of Ar-Raqqah (−0.20 unit, 95% CI −0.55 to 0.16).

**Impact by exposure to the conflict and social support.** In 2013 and 2015, participants who were directly exposed to the conflict did not report lower physical, mental, and social well-being, except for greater difficulties with access to food (OR = 1.36, 95% CI 1.10 to 1.67) and shelter (OR = 1.24, 95% CI 1.01 to 1.51; Supplementary Tables 5–7). Prior to the conflict, participants with social support consistently reported better physical, mental, and social well-being compared to participants without social support, adjusted for age and sex (Supplementary Fig. 3 and Supplementary Tables 8–10). However, the positive association between social support and well-being dissipated during the Syrian conflict (Supplementary Fig. 3 and Supplementary Tables 8–10). For example, Syrian participants with social support were 46% (OR = 1.46, 95% CI 1.22 to 1.75) more likely to be well-rested prior to the conflict, but this association weakened during the conflict (ratio of ORs = 0.50, 95% CI 0.15 to 0.85; relative excess risk due to interaction = −0.54, 95% CI −0.90 to −0.18). Sensitivity analyses based on complete-case analyses showed highly similar results (Supplementary Tables 7 and 10).

**Cross-national analyses with Syria and 162 other countries.** Descriptive statistics pooled across countries are presented in

**Table 1 Items measuring physical, mental and social well-being.**

| Items | Response options |
|---|---|
| **Physical well-being** | |
| *Health problem* | Yes/no |
| Do you have any health problems that prevent you from doing any of the things people your age normally can do? | |
| *Physical pain* | Yes/no |
| Did you experience the following feelings during a lot of the day yesterday? How about physical pain? | |
| *Well-restedness* | Yes/no |
| Did you feel well-rested yesterday? | |
| *Health care* | Satisfied/dissatisfied |
| In the city or area where you live, are you satisfied or dissatisfied with the availability of quality healthcare? | |
| *Food* | Yes/no |
| Have there been times in the past 12 months when you did not have enough money to buy food that you or your family needed? | |
| *Shelter* | Yes/no |
| Have there been times in the past 12 months when you did not have enough money to provide adequate shelter or housing for you and your family? | |
| **Mental well-being** | |
| *Negative emotions* | Yes/no |
| Did you experience the following feelings during a lot of the day yesterday? How about worry/sadness/stress/anger? (4 separate items) | |
| *Positive emotions* | Yes/no |
| Did you experience the following feelings during a lot of the day yesterday? How about enjoyment? Did you smile or laugh a lot yesterday? | |
| *Life satisfaction* | 0: worst possible |
| Please imagine a ladder with steps numbered from zero at the bottom to ten at the top. The top of the ladder represents the best possible life for you and the bottom of the ladder represents the worst possible life for you. On which step of the ladder would you say you personally feel you stand at this time? | 10: best possible |
| *Hope* | 0: worst possible |
| Please imagine a ladder with steps numbered from zero at the bottom to ten at the top. The top of the ladder represents the best possible life for you and the bottom of the ladder represents the worst possible life for you. Just your best guess, on which step do you think you will stand in the future, say about five years from now? | 10: best possible |
| **Social well-being** | |
| *Social support* | Yes/no |
| If you were in trouble, do you have relatives or friends you can count on to help you whenever you need them, or not? | |
| *Respect* | Yes/no |
| Were you treated with respect all day yesterday? | |
| *Freedom* | Satisfied/dissatisfied |
| In (this country), are you satisfied or dissatisfied with your freedom to choose what you do with your life? | |
| **Exposure status** | |
| Do you consider yourself an internally displaced person? | Yes/no |
| Since the start of the conflict in Syria over two years ago, have any of your immediate family members left the area they were living in and moved to (1) Somewhere else within the same governorate/ (2) To another governorate/ (3) Somewhere outside Syria? | Yes/no |
| Has someone from this household (1) lost their life/ (2) been injured as a result of the ongoing violence? | Yes/no |
| Has this household lost their main source of income due to the ongoing violence? | Yes/no |

*Note:* Bolded labels indicate different categories of variables, and the italic formatting is used to label each variable.

Supplementary Table 11. The declines in well-being in Syria reported in the serial cross-sectional individual-level analyses could be contextualised by comparing with the rest of the world (Supplementary Figs. 4 and 5, and Supplementary Tables 12–14). The multilevel analyses showed that prior to the conflict, Syria had similar levels of physical, mental, and social well-being compared to other regions (Fig. 3, Supplementary Fig. 4, and Supplementary Table 12).

From 2008 to 2015, all measures of well-being, aside from health problems, changed more substantially in Syria relative to the Eastern Mediterranean Region (Supplementary Fig. 4 and Supplementary Tables 12 and 13). For instance, difficulties with access to quality healthcare did not change in eastern Mediterranean (1.2%, 95% CI −5.7% to 3.3%) compared to a 31.7% (95% CI 11.7% to 51.6%) increase in Syria. The increase in negative emotions in Syria (38.6%, 95% CI 26.7% to 50.6%) was almost 10 times the change observed in the eastern Mediterranean (4.0%; 95% CI 1.5% to 6.5%). For social well-being, the absolute prevalence for the availability of social support decreased by 4.6%

(95% CI −7.5% to −1.7%) in the eastern Mediterranean, while the decrease in social support in Syria was more substantial with a 36.0% (95% CI −49.6% to −22.5%) decrease. These results held when comparing Syria to other regions.

We compared the changes in well-being in Syria with 162 other countries based on the random slopes from the multilevel models. For 11 out of 13 well-being indicators (with health problems and physical pain as the exceptions), the declines in well-being in Syria from 2008 to 2015 were the worst out of the 163 countries (Fig. 4 and Table 3).

We further examined the well-being trends in Syria in comparison with other countries that experienced war, major protests, and natural disasters. Consistent with previous analyses, all well-being indicators, except for health problems and physical pain, dropped more substantially in Syria than in other countries experiencing war, major protests, or natural disasters (Fig. 3, Supplementary Fig. 5, and Supplementary Table 14). For example, the proportion of participants satisfied with the availability of health care in Syria dropped 36.4% (95% CI

**Table 2 Descriptive statistics of 11,052 Syrian participants, Gallup World Poll 2008–2015.**

| Year | 2008 | 2009 | 2010 | 2011 | 2012 | 2013 | 2015 |
|---|---|---|---|---|---|---|---|
| **Demographic compositions** | | | | | | | |
| Sample size | 1209 | 2100 | 2035 | 2041 | 2043 | 1022 | 1002 |
| Women | 48.8% | 48.6% | 49.0% | 49.0% | 48.9% | 49.0% | 49.2% |
| Age, years (SD) | 32.8 | 33.4 | 34.0 | 33.7 | 34.0 | 33.8 | 33.5 |
| | (13.3) | (14.4) | (16.0) | (14.9) | (16.3) | (16.1) | (15.3) |
| Unemployed | — | 2.7% | 2.1% | 6.9% | 4.6% | 10.0% | 7.1% |
| Married | 55.4% | 54.1% | 64.1% | 55.5% | 56.0% | 53.5% | 54.8% |
| Elementary education or less | 64.9% | 49.2% | 61.5% | 61.1% | 62.1% | 61.8% | 61.0% |
| Household Income, thousands | — | 19.7 | 17.7 | 24.6 | 16.0 | 14.3 | 13.2 |
| (international $) (SD) | — | (26.7) | (25.8) | (33.9) | (18.2) | (19.4) | (22.7) |
| **Exposure status** | | | | | | | |
| % Internally displaced | — | — | — | — | — | 15.9% | 18.8% |
| % Family members displaced | — | — | — | — | — | 26.9% | 24.5% |
| % Someone from the household died | — | — | — | — | — | 0.8% | 4.0% |
| % Someone from the household injured | — | — | — | — | — | 1.8% | 7.7% |
| % Household lost source of income | — | — | — | — | — | 15.8% | 22.2% |
| **Physical well-being** | | | | | | | |
| Health problems compared to others | 25.6% | 23.6% | 11.8% | 11.7% | 10.8% | 15.4% | 17.2% |
| Physical pain | 38.3% | 32.6% | 16.1% | 26.1% | 16.0% | 14.5% | 15.0% |
| Well-rested | 56.7% | 65.9% | 59.9% | 66.4% | 63.8% | 38.0% | 36.3% |
| Satisfactory access to health care | 54.3% | 68.2% | 76.8% | 39.3% | 31.8% | 40.3% | 39.2% |
| Cannot afford food | 16.4% | 16.1% | — | 36.7% | 42.2% | 56.8% | 47.6% |
| Cannot afford shelter | 13.5% | 20.5% | 25.7% | 19.8% | 30.3% | 44.3% | 34.4% |
| **Mental well-being** | | | | | | | |
| Average life satisfaction, 0 to 10 (SD) | 5.32 | 4.98 | 4.46 | 4.04 | 3.16 | 2.69 | 3.46 |
| | (1.92) | (2.06) | (2.15) | (2.48) | (2.51) | (2.35) | (2.81) |
| Average hope, 0 to 10 (SD) | 6.32 | 6.34 | 5.75 | 5.15 | 5.38 | 5.63 | 4.88 |
| | (2.48) | (2.39) | (2.51) | (2.58) | (3.05) | (2.85) | (2.90) |
| Positive emotions | 62.7% | 60.4% | 51.1% | 65.5% | 47.5% | 38.2% | 36.0% |
| Negative emotions | 37.9% | 32.1% | 26.4% | 51.4% | 66.5% | 64.5% | 62.0% |
| **Social well-being** | | | | | | | |
| Social support | 71.2% | 84.2% | 93.4% | 57.6% | 58.8% | 58.5% | 46.4% |
| Respect | 80.9% | 91.7% | 87.6% | 78.9% | 68.2% | 60.9% | 57.4% |
| Freedom | 66.1% | 74.8% | 64.7% | 53.0% | 46.7% | 45.5% | 44.8% |

*Note*: Bolded labels indicate the categories of variables.

14.4% to 57.5%), 28.7% (95% CI 7.2% to 50.1%), and 35.0% (95% CI 13.9% to 56.2%) more compared to countries in armed conflict, social unrest and disasters, respectively. Negative emotions in Syria increased by 37.5% (95% CI 23.9% to 51.0%) more compared to countries in armed conflict, and social support decreased by 39.3% (95% CI 26.4% to 52.2%) more.

## Discussion

The current study takes a global perspective and examine the temporal patterns of physical, mental, and social well-being in war-torn Syria relative to the rest of the world. The population-representative sample of Syrian participants across multiple years before and during the conflict and the comprehensive examination of all three domains of health are significant improvement over previous studies[17,18]. Consistent with reports of rising mortality[8], blatant violations of international laws[29], and compromised safety for healthcare workers[13,14], the current study found that a wide spectrum of physical, mental, and social well-being has dropped in Syria during the conflict. However, the extent of the declines can be difficult to interpret without comparison (e.g., how large is a drop of 1.3 unit in hope?). We therefore provided the cross-national analyses to place the changes in well-being in Syria into global context. Notably, prior to the conflict, Syria's level of well-being was comparable to neighbouring countries. However, from 2006 to 2016, the substantial and pervasive declines in well-being in Syria were unmatched when compared to the eastern Mediterranean and other WHO regions. Changes in well-being in Syria are therefore not explained by the broader regional or global trends. Indeed, the fall in most well-being indicators in Syria has exceeded all countries in the Gallup World Poll, even in comparison to countries that have also suffered military conflict, civil strife, and natural disasters, including Haiti, Sudan, and Iran.

The magnitude of the average decline in Syria's life satisfaction —a drop of 1.1 SD—is striking compared to major life events such as bereavement and disability (typically associated with a drop in life satisfaction of 0.5 SD and 0.6 SD, respectively)[30,31]. When comparing the Syrian conflict to other major population events, the 2011 Fukushima disaster, characterised by a 9.1-magnitude earthquake, tsunami, and a nuclear plant meltdown, lowered life satisfaction by 0.12 SD for Japanese participants residing in the affected areas (compared to a drop of 1.1 SD in the Syrian conflict)[22]. Life satisfaction in Syria in 2013 was in fact the lowest recorded in the 2006–2016 Gallup World Poll. Importantly, 2015 was the last data collection in Syria by the Gallup World Poll. Since then, the Syria conflict grows in complexity, marked with escalations in international involvement but also the recapture of the city Raqqa from ISIS' control. Future research should continue to track Syrians' well-being through these major developments. The large psychological impact of the Syrian conflict reinforces the importance of scaling up evidence-based psychological interventions (e.g., Problem Management Plus developed by the WHO) in Syria and neighbouring countries with sizeable populations of Syrian refugees[32].

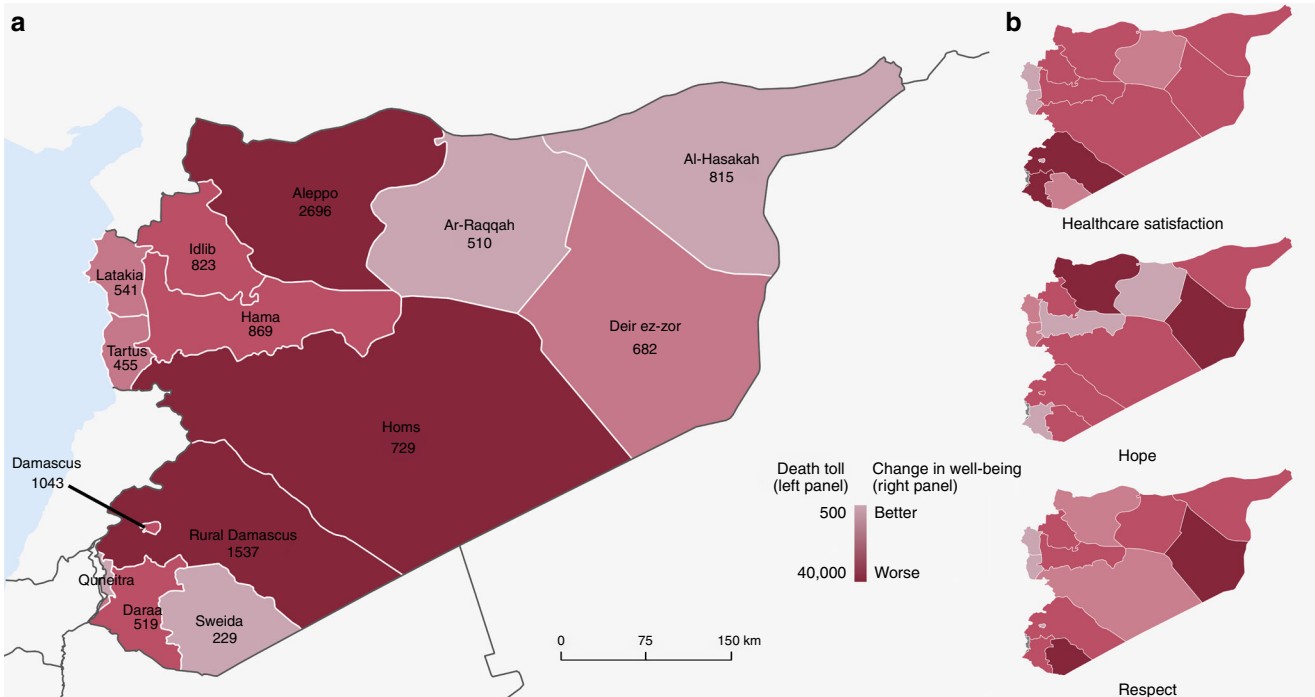

**Fig. 1 The geographical distribution of participants and well-being in Syria.** Panel **a** presents the number of participants in each governorate. Panel **b** presents the geographical differences in the declines in physical, mental, and social well-being from 2008 to 2015. This figure was created using QGIS developed by the QGIS Development Team (http://qgis.osgeo.org). Source data are provided as a Source Data file.

Both the degree of exposure to the Syrian conflict and social support were not associated with the levels of well-being during the Syrian conflict. This contrasts with the extensive disaster-related literature indicating the degree of exposure as the most important predictor of well-being, whereas the level of social support has consistently been shown to be protective[10–12]. Our finding suggests societal spillover effects, wherein well-being declined to a similar degree for participants with and without direct exposure to the conflict[10]. Moreover, the stress-buffering role of social ties may have ceased to function given the scale of the conflict in Syria[11,33].

Counter-intuitive improvements were observed for reported health problems compared to others and physical pain. These results may be partly explained by sampling bias and a 'pre-valence-induced concept change'. Specifically, recruiting a representative sample during major conflicts is extremely diffi-cult[19]. Although the Gallup used stratified random sampling to recruit participants that were representative in terms of age, sex and socioeconomic status, the samples might not have been truly representative in terms of direct exposure to the Syrian conflict. Due to safety concerns, extremely dangerous areas were excluded from the sampling process. Indeed, 19% of our sample were internally displaced compared to the national estimate of 33%, suggesting that the Syrian participants sampled might have been less exposed to the on-going conflict than the general population. This may therefore explain the unanticipated decline in reported health problems. Second, human judgement is context dependent, and recent research has shown 'prevalence-induced concept change' as a characteristic of human judgement[34]. When applied to the context of the Syrian conflict, prevalence-induced concept change can refer to the tendency to lowering the threshold for 'health' when fully healthy people become less prevalent. In other words, participants may not report relatively minor pain or health problems when people around them have lost their lives, experienced chemical attacks, or suffered severe injuries[29]. Pre-vious research has also documented that participants tended to

underreport health problems or pain when they expect their health to deteriorate[35,36]. The item on health problem also spe-cifically requires participants to compare themselves with others, providing further credence for this contextual effect. Future research should consider verifying and explaining these counter-intuitive results as underreports of health issues could delay proper medical treatments. Yet even among our sample of rela-tively physically well survivors, we observed dramatic declines in mental and social well-being. Moreover, we acknowledge that the sampling bias and prevalence-induced concept change, however unlikely, could have also biased the other 11 outcomes, and our observed changes in well-being might well be conservative estimates.

Some limitations of the current study should be noted. First, the current study examined temporal changes using a serial cross-sectional design. This precludes the assessment of within-person changes in well-being. However, repeated measurements of well-being measures within the same panel may result in system-atically lower scores, and thus the serial cross-sectional design avoids this panel conditioning bias[37]. Our findings demonstrated drastic changes in population average well-being, and future research should consider following Syrian participants over time to identify different trajectories of well-being change. Second, the physical, mental, and social well-being indicators analysed here were limited by the data availability in the Gallup World Poll, and the use of mostly single-item and dichotomous self-reported outcome measures is a limitation when compared to well-validated gold-standard measures of well-being. However, use of objective records to assess well-being was impeded by the wide-spread destruction of healthcare facilities, and indeed, the United Nations discontinued reporting the death tolls in the Syrian conflict, citing the lack of reliable sources[8,13,14,38]. Therefore, the current use of self-reported well-being measures covering topics related to healthcare, physical health, mental well-being, and social support complement past studies and provided useful, albeit imperfect, insights into life during the Syrian

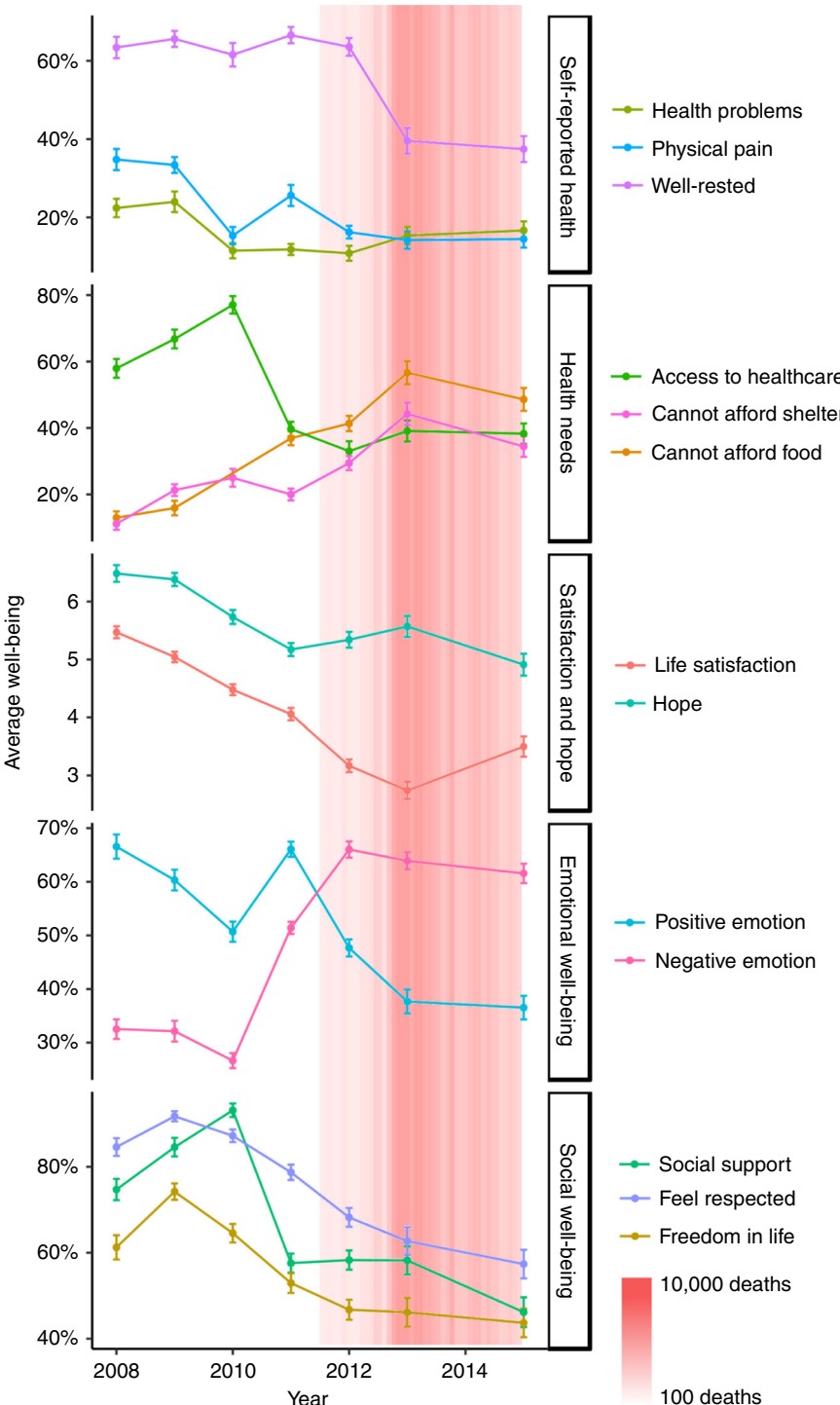

**Fig. 2 Overall trends in well-being in Syria from 2008 to 2015.** The shades of red represent the severity of the conflict based on monthly death toll data from the Syrian Center for Statistics and Research. Error bars indicate 95% CI (±1.96* standard error). This figure was created using R developed by the R Core Development Team (https://www.r-project.org/). Source data are provided as a Source Data file.

conflict. Future research should draw on well-validated measures to track Syrians' well-being over time. Third, although Gallup's use of a reasonably brief survey facilitates cross-temporal and cross-national comparisons, the vast cultural differences across the Globe could have affected the results. Fourth, the Gallup World Poll does not publicly publish its response rates. However, in a 2009 survey led by the Gallup in Arab countries, they achieved an adequate response rates (AAPOR RR1) of 59% in Syria[39]. The response rate during the conflict was likely lower, but

assuming a healthy responder effect, the sharp declines in well-being documented here may have nevertheless been overly conservative. Fifth, although military conflicts, major protests, and natural disasters can exact a substantial toll on physical, mental and social well-being[9,10,16], we did not detect large declines among affected countries in the Gallup World Poll. This might be due to the use of annual country-wide averages in well-being as the outcomes, which could have obscured acute drops in well-being in the most affected areas, especially when certain conflicts

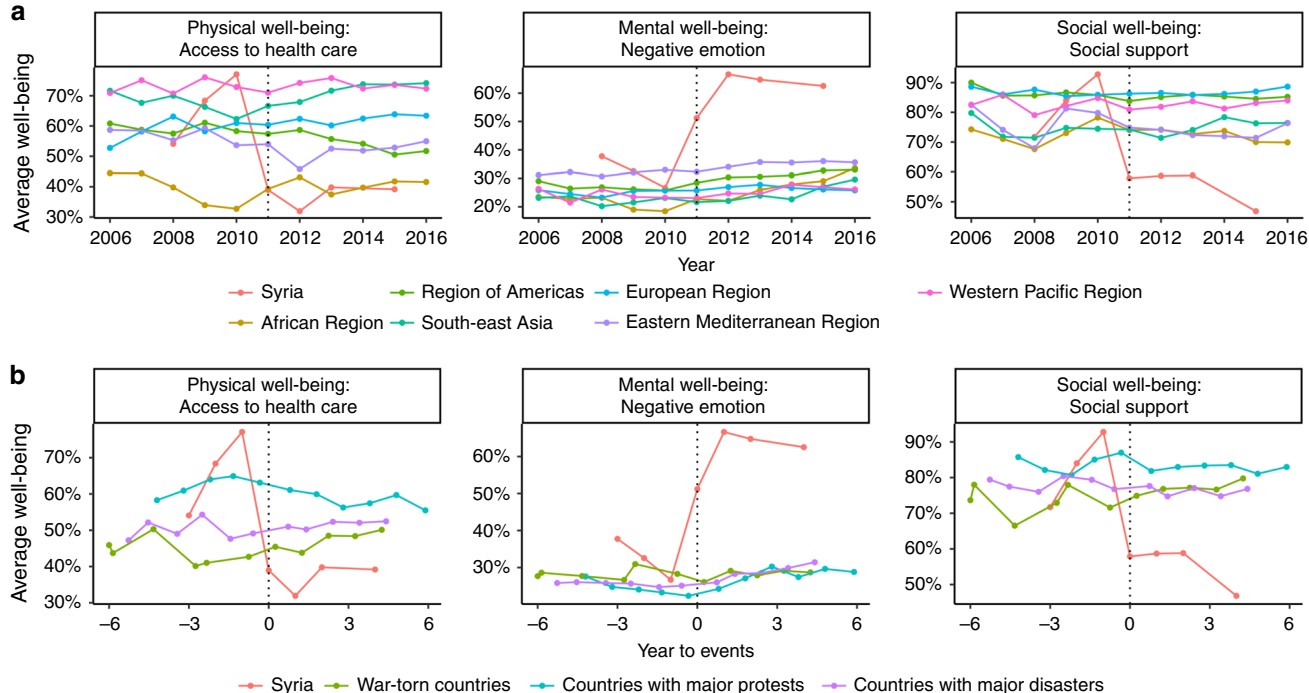

**Fig. 3 Temporal comparisons in physical, mental and social well-being.** Panel **a** compares the changes in Syria to other World Health Organization (WHO) regions. Panel **b** compares Syria to other countries that experienced war, protests, and natural disasters. The dotted line indicates the start of events. SI Appendix, Supplementary Fig. 4 and Supplementary Table 10, presents the trends for all 13 indicators. This figure was created using R developed by the R Core Development Team (https://www.r-project.org/). Source data are provided as a Source Data file.

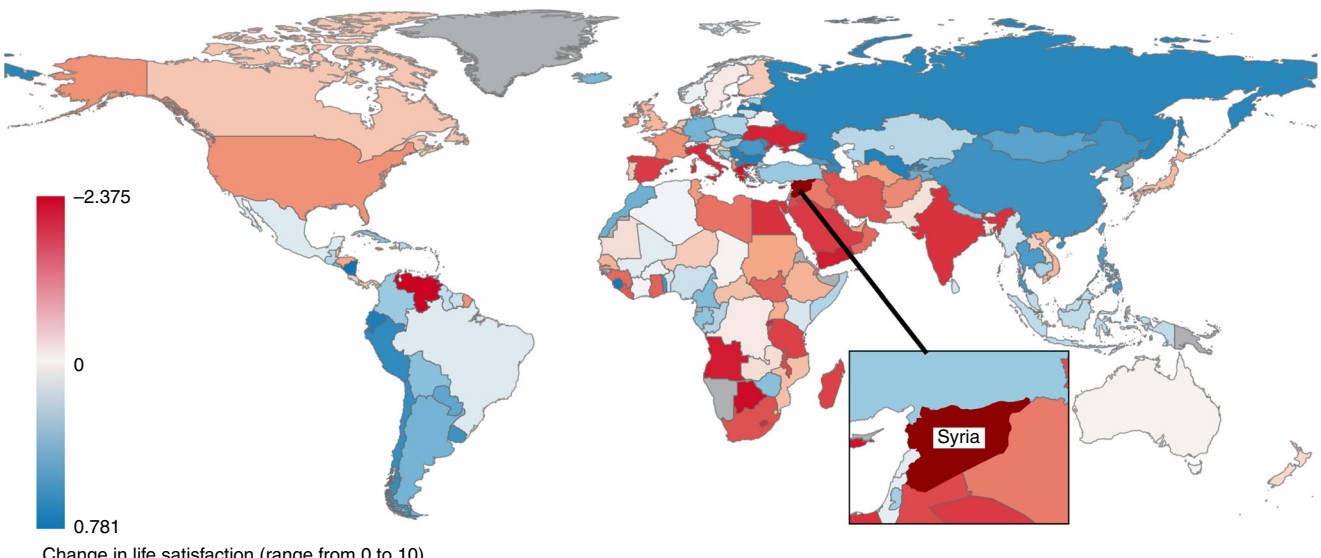

**Fig. 4 The temporal change in the level of life satisfaction in 163 countries from 2006 to 2016.** Syria's decline in life satisfaction is the largest among all surveyed countries. This figure was created using QGIS developed by the QGIS Development Team (http://qgis.osgeo.org). Source data are provided as a Source Data file.

and disasters in the study period are relatively localised. Nevertheless, the findings for Syria using the same outcomes indicate the overwhelming and enduring impact of the Syrian conflict on well-being.

Our findings show the dramatic and pervasive declines in physical, mental, and social well-being following the Syrian conflict. Recognising the urgency of humanitarian efforts to tend to the physical trauma resulted from the Syrian conflict, the less visible wounds to the psyche and social fabric are also far-reaching. Even in our sample of physically healthy and relatively less exposed Syrian participants, the psychological population impact of the Syrian conflict exceeded that of those experiencing bereavement, becoming disabled, and suffering from a major natural disaster. Our findings further reinforce the fundamental importance of peace for population health. To protect physical, mental, and social well-being, humanitarian efforts and targeted psychosocial interventions need to be underlined by an acceleration of the Syrian peace process. It is only through peace that we can rebuild social networks, address basic health needs, and restore hope for Syrians[40].

**Table 3 The top 5 countries with the greatest estimated declines in physical, mental, and social well-being from 2006 to 2016.**

| Rank | 1 | 2 | 3 | 4 | 5 |
|---|---|---|---|---|---|
| **Health problems** | Mozambique 10% | S. Sudan 9% | C.A.R 9% | Congo-Brazzaville 9% | Laos 8% |
| **Pain** | Laos 17% | Ivory Coast 14% | Chad 14% | Angola 13% | S. Sudan 13% |
| **Well-rested** | *Syria* −27% | Tunisia −9% | Egypt −6% | Pakistan −5% | Sudan −5% |
| **Health care** | *Syria* −32% | Venezuela −27% | Egypt −18% | Ghana −16% | Sudan −14% |
| **Cannot afford food** | *Syria* 41% | Nepal 31% | Iraq 27% | Laos 25% | Iran 23% |
| **Cannot afford shelter** | *Syria* 25% | Nepal 25% | Laos 23% | Mongolia 23% | Iraq 21% |
| **Life satisfaction** | *Syria* −2.4 | Venezuela −1.0 | Botswana −0.7 | C.A.R. −0.6 | Greece −0.6 |
| **Hope** | *Syria* −1.3 | Greece −0.9 | India −0.8 | Iran −0.8 | Liberia −0.7 |
| **Positive emotion** | *Syria* −27% | Egypt −7% | Tunisia −6% | Sudan −6% | Kuwait −5% |
| **Negative emotion** | *Syria* 39% | S. Sudan 14% | Zambia 13% | Uganda 13% | Gabon 12% |
| **Support** | *Syria* −36% | Somalia −14% | Albania −9% | Tunisia −8% | Iran −8% |
| **Respect** | *Syria* −34% | Niger −2% | Burkina Faso −1% | Ghana −1% | Mauritius −1% |
| **Freedom** | *Syria* −30% | Venezuela −15% | Sudan −12% | Afghanistan −12% | Mauritania −9% |

*Notes:* C.A.R = Central African Republic, S. Sudan = South Sudan
Bolded cells indicate row names and column header, and italic formatting is used to highlight Syria's ranking.

## Methods

**Data source and ethical approval.** This study was a secondary data analysis of de-identified data from the Gallup World Poll, which adopted a serial cross-sectional design, with multiple waves of data collection in Syria from 2008 to 2015 and 162 other countries from 2006 to 2016. The goal of the Gallup World Poll is to collect reliable data on attitudes, opinions, and well-being of randomly selected, population-representative samples across the world[25]. The inclusion criteria of the Gallup World Poll are non-institutionalised civilians aged 15 or above[25]. Annually, the Gallup World Poll surveys about 1000 participants in each society (Mean = 1187, SD = 844, range = 500 to 13,408) with a total sample of 1,722,558 participants across 163 territories (including Syria).

Random-digit-dialling telephone interviews were conducted in high-income countries (defined as where landline phones can reach at least 80% of the population)[25]. Face-to-face interviews were conducted in low-income countries, including Syria[25]. The samples were obtained using stratified random sampling, where primary sampling units (PSU) were sampled in subnational units (governorate in the case of Syria) such that sample sizes were proportionate to the populations of the subnational units[25]. After a PSU was sampled, a random-route method was used to select households, with at least 3 interview attempts[25]. Household members were randomly sampled using Kish grid[13]. Interviewers were provided training on administering the questionnaire and on executing the randomisation procedure with quality control from supervisors[25].

The current research involved secondary data analyses of de-identified data and did not require ethical approval at Washington University in St. Louis, Purdue University, and the University of Hong Kong where the analyses were conducted.

Face-to-face interviews in 13 out of 14 governorates of Syria were conducted by Gallup in 2008 (n = 1209), 2009 (n = 2100), 2010 (n = 2035), 2011 (n = 2041), 2012 (n = 2043), 2013 (n = 1022), and 2015 (n = 1002). The Quneitra governorate, which accounts for about 2% of the population, was not sampled in all waves[26]. In total, serial cross-sectional individual-level data from 11,452 participants was available.

The sample was not fully representative in terms of their exposure to the conflict due to safety concerns. In 2012, Homs, a governorate that was under siege by government forces since the early stage of the conflict, was excluded from sampling, and the data were representative of 93% of the Syrian population[26]. In 2013, Homs and about 25% of the selected PSUs in other governorates were replaced with other PSUs for security reasons, and the 2013 data were representative of 66% of the Syrian population[26]. In 2015, certain neighbourhoods within the Homs City and 30% of the PSUs in other governorates were replaced with safer PSUs, and the data from 2015 was representative of 68% of the Syrian population[26].

Cross-national data from the 2006–2016 Gallup World Poll were used to compare well-being of Syrian participants with the rest of the world. Data were aggregated at the national level with a total of 1435 country-years.

**Overview of measures included in Gallup World Poll.** Gallup has consulted with academic and intergovernmental institutions, such as the World Bank and United Nations, to select and validate the items included in the Gallup World Poll[25,41]. The questions included in the Gallup World Poll are described elsewhere and broadly related to business and economics, government and politics, and health[25]. Here, we based our analyses on virtually all the items in the Health and Well-being modules of the Gallup World Poll that were related to physical, mental, and social well-being. However, we excluded items in those modules that were not assessed before and during the Syrian conflict (e.g., questions about life engagement were only asked during the conflict with no pre-conflict data). We further selected items from the Food and Shelter and Citizen Engagement modules on affordability of food, affordability of shelter, social support, and perceived freedom because they pertained to physical and social well-being. Gallup showed that these measures demonstrated convergent and divergent validity with objective national-level statistics[25]. For example, the affordability of food and shelter are positively associated with GDP and negatively associated with mortality rate[25]. In addition, previous research has integrated many of the same measures in the current study as indicators of human welfare and subjective well-being[20,42–45]. To improve cross-national comparability and reduce cultural differences in response styles, a number of items were asked with only dichotomous response options (items presented verbatim in Table 1)[25].

**Exposure to the Syrian conflict.** Exposure to the Syrian conflict was assessed in 2013 and 2015 by asking participants whether they or their family members have been displaced, whether someone from the participants' household have been injured or have lost their life, and whether the household has lost its main source of income. We coded participants who reported 'Yes' to any of above conditions as having direct exposure to the Syrian conflict.

**Physical well-being measures.** Health problems were assessed by asking participants whether their health limited them in performing tasks that others of their same age can normally perform. Physical pain and well-restedness in the prior day were reported by participants. Satisfaction with availability of quality healthcare, access to food, and access to shelter were also assessed.

**Mental well-being measures**. Positive emotion was assessed with two items asking respondents to report whether they laughed or experienced enjoyment most of the time in the previous day. Negative emotion was assessed with four items by asking respondents whether they felt worried, sad, stressed, and angry most of the time in the previous day. Life satisfaction and hope for the next five years were assessed using the Cantril Ladder, which asked respondents to rate their lives on a 0 (worst possible) to 10 (best possible) scale. Previous research showed that single-item measures of mental well-being have satisfactory reliability and validity[46,47].

**Social well-being measures**. Social support was measured by asking respondents whether they have someone to count on. Perceived respect and satisfaction with freedom in life were also assessed.

**Classifications of countries**. In the national level analyses, countries were grouped into six WHO regions (African region, region of Americas, south-east Asian region, European region, eastern Mediterranean region, and western Pacific region). We compared well-being changes between Syria and the eastern Mediterranean region, as geopolitically they are more similar than other WHO regions[8].

Countries were also classified by occurrence of major population events. Countries with military conflicts were defined as having a death toll of 1000 or more due to military conflict in a calendar year based on the Uppsala Conflict Data Program, which is the largest and longest standing dataset on war[48]. Countries with major protests referred to countries with 800 or more reported protest-related arrests, injuries, and deaths based on a comprehensive review of global protests[49]. Countries with major natural disasters were defined as having a death toll of 1000 or more due to disasters in a calendar year based on the International Disasters Database, which compiles over 20,000 disasters across countries beginning in 1900 till present[50].

**Analyses of serial cross-section individual-level data**. Multilevel linear and logistic regression was used to examine the change in well-being in Syria and individual Syrian governorates. Each well-being measure was predicted from a linear time trend (0 = 2008; 0.14 = 2009; 0.29 = 2010; …; 1 = 2015) with random intercepts and random slopes for governorates. We used this coding scheme for the time trend to improve interpretability. With this coding scheme, the regression coefficient for the linear time trend can be interpreted as the change in well-being from 2008 to 2015. Individual responses were weighted to be representative of the Syrian population. Estimates for the time trends reflect the extent to which well-being in Syria changed from 2008 to 2015. The random slopes estimate the changes in well-being in each governorate. Age-specific and sex-specific trends of well-being were also examined by adding sex, age, and two-way and three-way interactions between age, sex, and time into the model.

We further compared the levels of physical, mental, and social well-being among participants by level of exposure to the Syrian conflict and social support. The comparison between participants with and without social support was evaluated by including an interaction term between social support and time, and we tested for both multiplicative and additive interaction. In the analyses on the buffering role of social support, social support was not included as an outcome. For analyses on the level of exposure, only data from 2013 and 2015 were included in the analyses as exposure was only assessed in these years.

Multilevel joint modelling multiple imputation based on Markov chain Monte Carlo algorithm was used to handle missing data in continuous and categorical variables while accounting for the clustering at the governorate level[51–53]. Overall, the percentage of complete data across all measures is 95.1% (i.e., 4.9% missing data; Supplementary Table 15 presents the missing data pattern). A total of 100 imputed datasets were created with 50,000 burn-in iterations and 5000 iterations between each imputed datasets. The long burn-in phase ensures that the imputation model converges before any imputed dataset was generated, and the large number of iterations between datasets ensures that the imputed datasets are independent from each other. Statistical estimates based on the 100 imputed datasets were combined using Rubin's rule[54]. We carried out sensitivity analyses with complete-case data. All statistical tests were two-sided.

**Analyses on national-level data**. We compared temporal changes in Syria's physical, mental, and social well-being with six WHO regions. Using national-level data from Syria and 162 countries, multilevel analyses were used to predict each well-being measure from survey year (−0.29 = 2006; −0.14 = 2007; 0 = 2008; 0.14 = 2009; …; 1 = 2015; 1.14 = 2016), dummy-coded variables for WHO regions (Syria as the reference group), and the interaction between survey year and regions. Random intercepts and random slopes for survey year at the country level were included.

In addition to the comparisons with the six WHO regions, we conducted a separate set of multilevel analyses to compare Syria with countries with military conflict, major protests, and natural disasters. Only countries with at least one year of data before and after the major population events were included in these analyses. To account for the different timing of the population events, we analysed by time to event instead of calendar year. All analyses were conducted using R version 3.3.3 statistical software (R Foundation for Statistical Computing, Vienna, Austria).

**Reporting summary**. Further information on research design is available in the Nature Research Reporting Summary linked to this article.

## Data availability

The data that support the findings of this study are available from Gallup but restrictions apply to the availability of these data, which were used under license for the current study. The data are not publicly available. Data are however available from the corresponding authors upon reasonable request and with permission of Gallup. A reporting summary for this Article is available as a Supplementary Information file. Source data are provided with this paper.

## Code availability

Codes are available upon request from the corresponding author.

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

## Acknowledgements

We thank Gallup for the data access and Ms. Betty Yuan, Ms. Yoona Kim, and Ms. Candi M.C. Leung for technical assistance. The monthly civilian death statistics were collected and distributed by the Syrian Center for Statistics and Research (CSR-SY; http://www.csr-sy.org/index.php?l=1). The CSR-SY is a non-governmental, non-profit organisation that was one of the primary sources for the United Nations on death toll-related statistics in Syria. The statistics used is available on the CSR-SY website (http://sn4hr.org/blog/category/report/monthly-reports/victims-death-toll/). This work was supported by a Seed Fund for Basic Research (#201709159014) awarded to FC and MN by the University of Hong Kong. The funders had no role in study design, data collection and analysis, decision to publish or preparation of the manuscript. We are especially grateful to the Gallup Organisation for collecting the Syrian data. This study on Syrians' well-being during the Syrian Civil War would not be possible without their data collection efforts.

## Author contributions

F.C. designed the study with input from all authors (A.K., L.T., E.D., J.J.J., R.E.L., M.N., G.M.L.). F.C. and A.K. were primarily responsible for conducting the national-level analyses. E.D. and F.C. were responsible for acquiring the access to the individual-level data on Syrian respondents. F.C. and L.T. were primarily responsible for designing and conducting the individual-level analyses. M.N. and G.M.L. provided critical feedback on the analytical strategies. M.N. and G.M.L. contributed equally. All authors participated in the interpretation of the results. F.C. drafted the manuscript with critical revisions from all authors. All authors (F.C., A.K., L.T., E.D., J.J.J., R.E.L., M.N., G.M.L.) approved of the final version of the manuscript.

## Competing interests

The authors declare no competing interests.
