## [Peer Review File · Nature Communications]

Reviewers' comments:

Reviewer #1 (Remarks to the Author):

In this paper, the authors present compelling evidence that the Syrian civil war has produced exceptional declines in well-being. I think this is a very important conclusion that needs to be shared with the scientific community. The authors make use of a very large and unique dataset from Gallup. The methods used in data collection have been developed carefully by this organization and its allied researchers, and the limitations (such as uneven sampling in especially unsafe areas of Syria) are clearly and openly acknowledged. I did not have any concerns regarding data analysis, although I should note that most of my research is experimental and I was not familiar with some of the statistical approaches used by the authors, so I would defer to other reviewers' evaluations of the analyses. That said, the author list includes some of the most well-equipped scholars in the world to analyze this data set.

Ideally, I would have liked to see a pre-registered analysis plan, but it seems plausible that the authors had too much familiarity with the relevant data to genuinely pre-register. Their analyses did appear quite comprehensive and transparent, and thus this is not a substantial concern.

The paper is clearly written and the conclusions are well-grounded in the data. Because this paper is so straightforward and well-done, I have just a few minor suggestions, which the authors can choose to utilize or ignore:

- 1) The graphs are almost impossible to interpret if printed in black and white. Perhaps this isn't a big problem given that the journal is online, but it would increase readability for those of us who still like paper if the graph formatting could be improved.
- 2) It seems to me that the people who collected the Syrian data in the midst of the civil war are the heroes of this research, and it feels a little odd to me that they are not mentioned by name, at least in the acknowledgements.
- 3) In the discussion on p. 14, the authors write that "from 2008 to 2015, the substantial and pervasive declines were unmatched..." but in Figure 3 and Table 2 the comparative data cover the years from 2006 to 2016, so I was puzzled by this apparent discrepancy in timeline.

Signed,

Elizabeth Dunn

Reviewer #2 (Remarks to the Author):

Thank you for the opportunity to review the manuscript “Physical, Mental, and Social Well-Being in Syria and 162 Countries from 2006 to 2016” for Nature Communication. I enjoyed this manuscript and found it to have many strengths. It was clearly written and well organized, and its topic is of great importance, given the scope and impact of the ongoing conflict in Syria. The authors also drew on rich data for the analysis, and I appreciated that they compared well-being in Syria to all of the other 162 countries, countries within specific regions, and countries that have experienced various forms of unrest in recent years. Analyses were weighted to be representative of the Syrian population, missing data were appropriately and impressively handled, and a sensitivity analysis of complete cases was conducted. The figures were extremely useful in illustrating the descriptive data. The Discussion was very strong, particularly the section on limitations and their potential influence on patterns of results.

Despite these strengths, I had several suggestions for how the manuscript as a whole could be further developed, as well as how the methods and analyses could be clarified. Overall, I see these suggestions as relatively minor and believe that they could be addressed through revision. I outline the considerations by section below.

Title/Abstract

- 1) Very minor comment: I would recommend that the title read “162 other countries” rather than “162 countries” for clarity.
- 2) The authors might consider including the dates that data were collected in the Abstract, else the reader might be confused by how the title lists “2006 to 2016” whereas a finding in the Abstract lists “2008 to 2015.”

Introduction

3) In general, I found the Introduction to be clearly written and well organized, but strikingly short. Upon my first reading, I assumed that this must be due to a short word limit, but then reviewed the journal’s Author Guidelines and saw that the limit is actually much longer than the manuscript in its current form. I would therefore encourage the authors to further develop the Introduction to better situate the study in the extant literature on the long-term impact of various disasters on wellbeing. Some questions that the authors might consider in developing their review: Have other studies drawn on representative archival data to explore pre to post-disaster changes? If so, have they been able to look at these three dimensions of wellbeing? Have they compared changes in wellbeing in an affected country to other countries, both in general and others affected by similar crises? What aspects of disaster exposure predicted wellbeing in prior literature? How exactly have exposure and social support been found to interact in predicting outcomes? Addressing these questions would help illustrate the study’s unique contribution.

Method

4) The use of two different data sources – the first, from Syria annually from 2008 to 2015, and the second, from Syria and 162 additional countries annually from 2006 to 2016 – seemed a strength, but it was often confusing to me as a reader which data were used for which analysis. One possible means of

clarification could be to consistently label each data source (e.g., Syria National Data; Cross-National Data) throughout.

5) Is Homs a governorate? Please clarify for readers who are unfamiliar with Syrian geography.

6) Which measures were asked in the Syrian national data (2008-2015) vs. the cross-national data (2006-2016)?

7) How was exposure included in the analysis? Was each aspect (displacement, injury, income loss) looked at separately? Or were they combined in some way? Was data from both 2013 and 2015 used in the analysis? Greater clarity is needed here.

8) A bit more detail on the well-being measures would be helpful. Have these measures been used in prior research? Is there any empirical evidence of their validity and other psychometric properties?

Statistical Analysis

9) I would advise against labeling the analysis of Syria national data “individual-level.” To me, this suggests a longitudinal design, whereas my understanding is that this was a serial cross-sectional data.

10) A linear time trend was used such that 0=2018 and 1=2015. Was only the data from 2008 and 2015 used? If so, did the authors consider an alternative analysis including all years? If not, how were other years coded?

11) I appreciated that the investigators looked at age- and sex-specific trends. However, this is another aspect for which I would have liked to see some background information in the Introduction. What does previous research say about age and sex differences in responses to disasters? Also, how was age coded in the analysis?

12) Greater clarity about the nature of the additive and multiplicative interactions would be useful. Was social support interacting with exposure? If so, which aspect of exposure? Or, was social support interacting with time? Please clarify. Also, was social support excluded among the dependent variables for these analyses? (Note: from the supplementary information, it seems that support was interacting with time, and that social support was indeed excluded. I would recommend clarifying these issues in the main document. Did the authors consider assessing interactions between exposure and social support?).

13) Again, I thought that the authors’ handling of missing data was a strength. However, I would have liked to see descriptive data on the extent of missing data (range of % missing for variables in the analysis; number and percentage of complete cases).

14) For the cross-national analyses, I again wondered about the years of data included. Was it again only 2008 and 2015? I assumed that this was to be consistent with the Syria-only analysis, but some justification would be useful, given that data were available from 2006 to 2016.

Results

15) Overall, I found the Results to be clearly written. One aspect that was confusing was that the analysis of the social support interactions seemed to be written as if there were longitudinal participant data (“the decline in well-restedness was sharper for participants with social support...”). This should be revised to be consistent with a serial cross-sectional design.

16) Did the results of the complete cases sensitivity analyses differ at all from the analyses with multiple

imputation?

Discussion

17) Can the authors define “societal spillover effects” and provide a citation for this phenomenon?

18) Can the authors define and provide a citation for “wartime contextual effect”?

19) The authors might consider including the use of single items, mostly dichotomous, among the limitations. Although these did facilitate cross-cultural comparisons, future research should draw on more in-depth measures with stronger psychometric properties, particularly for the within-country analysis.

Supplemental

20) Table S4 – Were ORs excluded for dependent variables for which the omnibus test was non-significant? I would recommend clarifying in the table note. Also, can confidence intervals be added to this table?

21) Table S11 – I would recommend also listing in bold when Syria was among the *top* five countries on negative indicators (e.g., health problems, physical pain).

Reviewer #3 (Remarks to the Author):

Title Manuscript: Physical, mental, and social well-being in Syria and 162 countries from 2006 to 2016

Overall

This is a very important and well-written manuscript, that describes the social and health effects of the Syrian war on the Syrian population in relation to other conflict and disaster-affected areas in the world. The authors used the data of the Gallup World Poll, a poll that includes Syrian samples (with Ns of 1002 to 2100 varying between waves) in which data such as physical protection/ shelter, access to health care and well-being were measured at several intervals between 2008-2015. To this reviewer’s knowledge, these data have not been used for this purpose before, and they provide a very important insight in the situation of the people within Syria in these extremely adverse times.

Abstract

It would be correct to describe in the abstract that the Gallup World Poll data were used, in order to facilitate researchers when systematically searching for studies using this dataset.

It is stated that “Syrians regardless of the degree of exposure were similarly affected”. This is a rather bold statement since only a few exposures were measured. No data are available on sexual violence (information which Syrians may not easily disclose anyway due to stigma), and on more subtle forms of exposure to trauma with high impact such as witnessing atrocities.

Introduction

The introduction is adequate, but would certainly benefit from a more grounded presentation of the variables studied. Why were variables such as social support selected, or access to health care, and for instance economic or financial circumstances were not? A more balanced overview of the vast literature on health and well-being in response to war and crisis should be presented, that should have guided the authors in deciding which variables to select.

Results

Table 1 shows that some variables do not decline as dramatically as one would expect within these circumstances. An example is hope, which drops only 1 point on a scale of 0 to 10 from pre-conflict to during the conflict.

Interestingly, “health problems compared to others” already dropped before the start of the conflict, in 2010, and it did not increase back to the pre-conflict level in the period from 2011-2015. How would the others explain this? Did other changes occur in the Syrian society that could explain this result?

A comparison has been done with 162 other countries. In the methods (under “Classification of countries”) it is stated that there was a focus on comparisons between Syria and other countries in the Eastern Mediterranean, but this is not reflected in the results.

Discussion

The crisis in Syria started in 2011. From 2015, the Syrian conflict became an international conflict, involving other countries including Turkey and Russia. This is also the year in which the refugee streams to Europe were at its peak. This was also the year in which the last measurement wave was carried out. It would be interesting if the authors would reflect more in-depth on how the timing of the assessments in relation to the political situation on the ground may have affected the results. In addition, the situation has further evolved after 2015, do the authors expect the same outcomes if they would have included more recent data?

Another issue that would deserve further elaboration and discussion, is whether the results differ between regions in Syria. Regions that may have been under ISIS control may be differently affected than regions under the control of the Syrian government. This might have had a direct effect on important variables such as access to health care.

The surprising finding that some variable improved instead of worsened, has been explained by the authors by a sampling bias and “war time contextual bias” (the latter being a form of a response shift or recalibration). However, the sampling bias would have affected all variables measured, and as such, this explanation undermines all other effects found in this study, also the intuitive ones. Thus, the authors should do a better job in explaining these specific improvements, or lack of clear deterioration. As stated earlier, are other factors present that may explain why self-reported health problems are not increasing? And perhaps reported pain was not decreasing since Syrians may have still had still access to pain medication, despite the reduced access to health care. It is commonly known that Syrians, even when displaced to other countries, still have contact with Syrian doctors, who may prescribe medications. It would be worthwhile to include a Syrian professional in the author team, who might provide a more informed and in-depth understanding of the specific effects found. Or the results could

have been presented to a focus group of Syrian (refugee) stakeholders, who might provide further clues to the interpretation. Such a mixed-methods approach would certainly benefit the paper.

A further issue to discuss is whether cultural differences between samples might have affected the results, specifically the between-country comparisons. Are the authors confident that variables such as health problems and well-being can be reliably and validly measured across cultural settings with the same questions? This might be a potential limitation, especially when comparing countries.

Access to health care is a very salient variable in this study, dropping dramatically from 2011 onwards. The authors conclude that along with providing accelerating the peace process, psychosocial interventions should be offered to Syrians affected. It will be challenging to provide psychosocial interventions to Syrians when the war is ongoing, but efforts are made to train non-specialist helpers in delivering mental intervention in low resource areas where access to health care is extremely limited. An example is the scaling-up of the evidence-based WHO PM+ programme across Syrians in Europe and the Middle East (e.g., Sijbrandij et al., Strengthening mental health care systems for Syrian refugees in Europe and the Middle East: integrating scalable psychological interventions in eight countries, *Eur J Psychotraumatology*, 2017, 8 (2))

Methods

The Gallup dataset may be more extensively described (how it is collected, for what purpose, by whom, etc.)

The authors state that since the data were secondary data, no ethics approval was obtained from accredited ethics review boards in St Louis or Hong Kong. Nevertheless, the authors should provide a formal waiver of at least one of these committees.

More information should be provided on how physical wellbeing was measured. Which the questions were asked, and on which scale were the responses scored?

Reviewers' comments:

Reviewer #1 (Remarks to the Author):

In this paper, the authors present compelling evidence that the Syrian civil war has produced exceptional declines in well-being. I think this is a very important conclusion that needs to be shared with the scientific community. The authors make use of a very large and unique dataset from Gallup. The methods used in data collection have been developed carefully by this organization and its allied researchers, and the limitations (such as uneven sampling in especially unsafe areas of Syria) are clearly and openly acknowledged. I did not have any concerns regarding data analysis, although I should note that most of my research is experimental and I was not familiar with some of the statistical approaches used by the authors, so I would defer to other reviewers' evaluations of the analyses. That said, the author list includes some of the most well-equipped scholars in the world to analyze this data set.

Response: Thank you for the comments!

Ideally, I would have liked to see a pre-registered analysis plan, but it seems plausible that the authors had too much familiarity with the relevant data to genuinely pre-register. Their analyses did appear quite comprehensive and transparent, and thus this is not a substantial concern.

Response: We understand and share the reviewer's view towards pre-registration. Pre-registration is becoming more popular in experimental research, but the guidelines for secondary analyses are less clear, partly for reasons that the reviewer pointed out (e.g., familiarity with the dataset). Please note that the current study started in 2017, and we regrettably could not benefit from a recent paper on promoting the transparency of analyses of existing data (Weston et al., in press, *Advances in Methods and Practices in Psychological Science*).

Weston SJ, Ritchie SJ, Rohrer JM, Przybylski AK. Recommendations for Increasing the Transparency of Analysis of Preexisting Data Sets. *Advances in Methods and Practices in Psychological Science*. Forthcoming 2018.

The paper is clearly written and the conclusions are well-grounded in the data. Because this paper is so straightforward and well-done, I have just a few minor suggestions, which the authors can choose to utilize or ignore:

1) The graphs are almost impossible to interpret if printed in black and white.

Perhaps this isn't a big problem given that the journal is online, but it would increase readability for those of us who still like paper if the graph formatting could be improved.

Response: Thank you for the feedback. We added different line types and shapes to Figures 2 and 3 to help differentiate the outcomes in black and white.

2) It seems to me that the people who collected the Syrian data in the midst of the civil war are the heroes of this research, and it feels a little odd to me that they are not mentioned by name, at least in the acknowledgements.

Response: The Syrian data were collected by Gallup and its local partners, and we very much agree with the reviewer about the indispensable role they played in this research. However, we do not have access to the names of specific individuals involved in the data collection. We added the following to the acknowledgement: "We are especially grateful to the Gallup Organization for collecting the Syrian data. The current study on Syrians' well-being during the Syrian Civil War would not be possible without their data collection efforts." (p. 29)

3) In the discussion on p. 14, the authors write that "from 2008 to 2015, the substantial and pervasive declines were unmatched..." but in Figure 3 and Table 2 the comparative data cover the years from 2006 to 2016, so I was puzzled by this apparent discrepancy in timeline.

Response: The data from the Gallup World Poll covered 2006 to 2016, but not all countries were sampled in all years. Data collection in Syria took place from 2008 to 2015. For the cross-national sample, we included data from 2006, 2007, and 2016 to improve the estimation of well-being secular trends in the other 162 countries. We agree with the reviewer's point about clarity, and accordingly changed the wording in the discussion section to "from 2006 to 2016" (p.17).

Reviewer #2 (Remarks to the Author):

Thank you for the opportunity to review the manuscript "Physical, Mental, and Social Well-Being in Syria and 162 Countries from 2006 to 2016" for Nature Communication. I enjoyed this manuscript and found it to have many strengths. It was clearly written and well organized, and its topic is of great importance, given the scope and impact of the ongoing conflict in Syria. The authors also drew on rich data for the analysis, and I appreciated that the compared well-being in Syria to all of the other 162 countries, countries within specific regions, and countries that have experienced various forms of unrest in recent years. Analyses

were weighted to be representative of the Syrian population, missing data were appropriately and impressively handled, and a sensitivity analysis of complete cases was conducted. The figures were extremely useful in illustrating the descriptive data. The Discussion was very strong, particularly the section on limitations and their potential influence on patterns of results.

Response: Thank you very much!

Despite these strengths, I had several suggestions for how the manuscript as a whole could be further developed, as well as how the methods and analyses could be clarified. Overall, I see these suggestions as relatively minor and believe that they could be address through revision. I outline the considerations by section below.

Title/Abstract

1) Very minor comment: I would recommend that the title read “162 other countries” rather than “162 countries” for clarity.

Response: We have edited the title accordingly.

2) The authors might consider including the dates that data were collected in the Abstract, else the reader might be confused by how the title lists “2006 to 2016” whereas a finding in the Abstract lists “2008 to 2015.”

Response: We edited the abstract based on this reviewer’s and Reviewer 1’s feedback above. See also our response 3) to Reviewer 1.

Introduction

3) In general, I found the Introduction to be clearly written and well organized, but strikingly short. Upon my first reading, I assumed that this must be due to a short word limit, but then reviewed the journal’s Author Guidelines and saw that the limit is actually much longer than the manuscript in its current form. I would therefore encourage the authors to further develop the Introduction to better situate the study in the extant literature on the long-term impact of various disasters on wellbeing. Some questions that the authors might consider in developing their review: Have other studies drawn on representative archival data on explore pre to post-disaster changes? If so, have they been able to look at these three dimensions of wellbeing? Have they compared changes in wellbeing in an affected country to other countries, both in general and others affected by similar crises? What aspects of disaster exposure predicted wellbeing in prior literature? How exactly have exposure and social support been found to interact in predicting outcomes? Addressing these questions would help illustrate the study’s unique contribution.

Response: Thank you for the suggestion to further develop the introduction. We have elaborated on the questions that this reviewer so helpfully provided. The revision seeks to highlight the three dimensions of well-being and introduce the cross-national comparisons by referencing prior literature on the long-term impact of various disasters, as follows (p. 3-4 in manuscript):-

“The Syrian conflict has widespread consequences on civilian health and well-being beyond mortality and displacement. Due to widespread destruction of health care facilities, progress in infant mortality has reversed, infectious diseases have been on the rise, and patients with chronic diseases have lost access to treatment. The damage to residential buildings and food distribution systems disrupt the basic needs for shelter and food. Exposure to war-related brutality is a salient risk factor for debilitating psychological conditions, with long-term impact that usually persists beyond the course of the war. Armed conflicts also destroy families and communities and force civilians into displacement, thus severely compromising social capital. Based on prior findings from major disasters, younger age, being female, exposure to the conflict, and lack of social support are additional risk factors for well-being decrement. However, objective assessment of population well-being is obstructed by the unprecedented “weaponisation of health care” (i.e. the denial of access to health care as a strategy of war). Therefore, Syrians’ self-reported well-being could be particularly relevant to measuring an important aspect of the impact of the conflict. Based on the World Health Organization’s holistic definition of health, we sought to assess the changes in self-reported physical, mental, and social well-being and to characterize well-being by sex, age, social support, and exposure to the conflict.

Although a large corpus of research has focused on individual major population events, comparison across different events is rarely conducted. Notably, in mental well-being research, studies have found little to no long-term decrease following the 2007-2008 global financial crisis, 2011 Fukushima disaster, or 2015 Paris terrorist attack. These findings give evidentiary support to an influential hypothesis in the psychology literature that life circumstances may play a limited role in changing long-term well-being. However, these findings do not appear to be consistent with empirical research on disaster and health. To clarify such apparent divergence, we adopted a global perspective to evaluate the changes in well-being in Syria relative to the rest of the world, including comparisons with other countries’ experiences with armed conflict, social unrest, and natural disasters.”

Method

4) The use of two different data sources – the first, from Syria annually from 2008 to 2015, and the second, from Syria and 162 additional countries annually from 2006 to 2016 – seemed a strength, but it was often confusing to me as a reader which data

were used for which analysis. One possible means of clarification could be to consistently label each data source (e.g., Syria National Data; Cross-National Data) throughout.

Response: To clarify, all data used in the current study come from the Gallup World Poll, and the data from Syria is a subset of the Gallup World Poll. We thank the reviewer for the suggestion to label the data consistently. Here we also took into account comment #9 (which recommended us to clearly describe the Syrian data as serial cross-sectional). We incorporated the reviewer's suggestion and have now consistently labeled the data as 'serial cross-sectional individual-level data' and 'cross-national data' throughout the manuscript. In addition, we revised the beginning of the method section to include a paragraph describing the Gallup World Poll (p. 21-22). This should help clarify the data source.

5) Is Homs a governorate? Please clarify for readers who are unfamiliar with Syrian geography.

Response: The reviewer is correct that Homs is a governorate. We have clarified this on p.23.

6) Which measures were asked in the Syrian national data (2008-2015) vs. the cross-national data (2006-2016)?

Response: We thank the reviewer for the comment, and to echo earlier responses, we should have been clearer that the Syrian data and the cross-national data both come from the Gallup World Poll. Therefore, they share many 'core' measures, which the Gallup World Poll has regularly assessed across countries. Reference 24 and 39 of the manuscript provide additional details about these core questions. A relatively accessible (but less formalized) introduction to the measures can be found in the following website:

https://media.gallup.com/dataviz/www/WP_Questions_WHITE.pdf

Broadly speaking, the ~100 core questions are grouped under different domains (e.g., citizen engagement; law and order; work and business). In the current study, we focus mostly on measures under the 'well-being' module. We selected the measures based on their relevance to the World Health Organization's holistic definition of health as physical, mental, and social well-being.

In addition to these core questions, certain country- and year-specific items are added to the surveys. For example, the questions on exposure to the Syrian conflict (e.g., internal displacement) were only asked in Syria in 2013 and 2015.

As reviewers 2 and 3 both commented on the measures used in the current study, we moved the original Table S1 (which presents the items verbatim) to the main text as Table 1 (p. 5-6).

7) How was exposure included in the analysis? Was each aspect (displacement, injury, income loss) looked at separately? Or were they combined in some way? Was data from both 2013 and 2015 used in the analysis? Greater clarity is needed here.

Response: We included the questionnaire items for exposure in Table 3 in the revised manuscript. To elaborate, exposure status was assessed with the same 7 items in 2013 and 2015:

- 1) Do you consider yourself an internally displaced person? That is, you had to leave your home for any reason and now live in a different part of Syria?
- 2) Since the start of the conflict in Syria over two years ago, have any of your immediate family members left the area they were living in and moved to any of the following areas? Somewhere else within the same governorate.
- 3) Since the start of the conflict in Syria over two years ago, have any of your immediate family members left the area they were living in and moved to any of the following areas? To another governorate.
- 4) Since the start of the conflict in Syria over two years ago, have any of your immediate family members left the area they were living in and moved to any of the following areas? Somewhere outside Syria.
- 5) Has someone from this household lost their life as a result of the ongoing violence?
- 6) Has someone from this household been injured as a result of the ongoing violence?
- 7) Has this household lost their main source of income due to the ongoing violence?

In our analysis, we coded participants who answered 'Yes' to any of the 7 questions as having direct exposure to the Syrian conflict. Data from both 2013 and 2015 were used in the analysis with year included as a covariate. Based on the reviewer's feedback, we have clarified the coding strategy in the method section.

"We coded participants who reported 'Yes' to any of above conditions as having direct exposure to the Syrian conflict." (p. 24)

8) A bit more detail on the well-being measures would be helpful. Have these measures been used in prior research? Is there any empirical evidence of their validity and other psychometric properties?

Response: Thank you for the opportunity to clarify the details of the well-being measures. The mental well-being outcomes (which are often referred to as subjective well-being in psychology) have been extensively studied. Previous research showed that single-item measures of subjective well-being have satisfactory reliability and validity (Cheung & Lucas, 2014, *Quality of Life Research*; Lucas & Donnellan, 2012, *Journal of Research in Personality*). As examples, the specific mental well-being measures within the Gallup World Poll have been used to study 1) the well-being impact of the 2008 Economic Recession (Deaton, 2011, *Oxford Economic Papers*), 2) the association between income inequality and mental well-being (Ng & Diener, 2019, *Applied Research in Quality of Life*), and 3) thresholds for income satiation (Jebb, Tay, Diener, & Oishi, 2018, *Nature Human Behaviour*; Kahneman & Deaton, 2010, *PNAS*).

For the physical and social well-being measures, Reference 24 refers to a document prepared by Gallup on the methodology of the Gallup World Poll. It includes evidence that these well-being measures show convergent and divergent validity with objective national-level statistics. For example, the affordability of food and shelter are positively associated with GDP and negatively associated with mortality rate. In addition, previous research has integrated many of the same measures in the current study as indicators of human welfare (Diener & Tay, 2015, *International Journal of Psychology*). We have incorporated these additional references to the method section (p. 24).

Cheung F, Lucas RE. Assessing the validity of single-item life satisfaction measures: Results from three large samples. *Quality of Life research*. 2014 Dec 1;23(10):2809-18.

Deaton A. The financial crisis and the well-being of Americans: 2011 OEP Hicks Lecture. *Oxford economic papers*. 2012;64(1):1-26.

Diener E, Tay L. Subjective well-being and human welfare around the world as reflected in the Gallup World Poll. *International Journal of Psychology*. 2015 Mar;50(2):135-49.

Jebb AT, Tay L, Diener E, Oishi S. Happiness, income satiation and turning points around the world. *Nature Human Behaviour*. 2018 Jan;2(1):33.

Kahneman D, Deaton A. High income improves evaluation of life but not emotional well-being. *Proceedings of the national academy of sciences*. 2010 Sep 21;107(38):16489-93.

Lucas RE, Donnellan MB. Estimating the reliability of single-item life satisfaction measures: Results from four national panel studies. *Social Indicators Research*. 2012 Feb 1;105(3):323-31.

Ng W, Diener E. Affluence and Subjective Well-Being: Does Income Inequality Moderate their Associations?. *Applied Research in Quality of Life*. 2019 Mar 15;14(1):155-

70.

Statistical Analysis

9) I would advise against labeling the analysis of Syria national data “individual-level.” To me, this suggests a longitudinal design, whereas my understanding is that this was a serial cross-sectional data.

Response: Based on the reviewer’s feedback, we labelled these analyses as “Analyses of serial cross-sectional individual-level data” (p. 7). This should make clear that we did not use a longitudinal design and distinguishes this set of individual-level analyses from the national-level analyses that came later in the manuscript.

10) A linear time trend was used such that 0=2008 and 1=2015. Was only the data from 2008 and 2015 used? If so, did the authors consider an alternative analysis including all years? If not, how were other years coded?

Response: All data available between 2008 and 2015 were included in the analyses. The linear time trend was coded such that ‘0’=‘2008’, ‘0.14’ (1/7) = ‘2009’, ‘0.29’ (2/7) = ‘2010’,..., ‘1’=‘2015’. We scaled the trend this way to improve interpretability. The regression coefficient for the time trend can be interpreted as the total change in the outcome from 2008 to 2015 (i.e., as time increases by 1 unit). We have clarified this point in the manuscript on p. 26.

11) I appreciated that the investigators looked at age- and sex-specific trends. However, this is another aspect for which I would have liked to see some background information in the Introduction. What does previous research say about age and sex differences in responses to disasters? Also, how was age coded in the analysis?

Response: Prior review found that female gender and young age are risk factors for poorer well-being. We have added reference to this literature in our introduction (p. 3). Age was self-reported by participants, and it was grand-mean centered before entering into the analysis. For Supplementary Figure 2, to simplify data visualization, we coded participants as younger adults (aged 35 or below), mid-life adults (aged 36-55), and older adults (56 or older).

Goldmann E, Galea S. Mental health consequences of disasters. Annual review of public health. 2014 Mar 18;35:169-83.

12) Greater clarity about the nature of the additive and multiplicative interactions would be useful. Was social support interacting with exposure? If so, which aspect of

exposure? Or, was social support interacting with time? Please clarify. Also, was social support excluded among the dependent variables for these analyses? (Note: from the supplementary information, it seems that support was interacting with time, and that social support was indeed excluded. I would recommend clarifying these issues in the main document. Did the authors consider assessing interactions between exposure and social support?).

Response: The reviewer is correct that social support was interacting with time, and support is not included as an outcome in these analyses. We agree with the reviewer’s assessment and clarified these issues in the main text.

“The comparison between participants with and without social support was evaluated by including an interaction term between social support and time, and we tested for both multiplicative and additive interaction. In the analyses on the buffering role of social support, social support was not included as an outcome.” (p. 27)

The reviewer also inquired about the possibility of interactions between exposure and social support. We believe the hypothesis for testing this interaction is that social support may buffer the impact of exposure to the conflict. We note that in the analyses that we presented, the regression coefficients for exposure were non-significant for 11 out of 13 outcomes. Because we did not find consistent well-being differences between participants with and without exposure, it would be difficult to test the buffering role of social support.

Nevertheless, as the data are proprietary and access is limited, we analysed the data to provide additional information for the reviewer to evaluate our manuscript. The table below presents the OR and 95% CIs for the interaction terms between social support and exposure. One out of the 12 interactions (positive emotion) was statistically significant, but the direction of the interaction is opposite to what one might expect (the association between exposure and positive emotion is more negative for participants with social support). We did not include this table in the revised manuscript.

Outcome	OR	95% CIs
Health Problem	1.181	[0.701, 1.989]
Pain	1.062	[0.604, 1.870]
Well-rested	0.889	[0.571, 1.383]
Health Care	0.773	[0.509, 1.174]
Cannot Afford Food	1.353	[0.872, 2.099]
Cannot Afford Shelter	1.199	[0.779, 1.846]
Respect	1.176	[0.765, 1.807]
Freedom	1.116	[0.734, 1.696]
	b	95% CIs
Life Satisfaction	-0.305	[-0.810, 0.201]

Hope	0.226	[-0.355, 0.807]
Positive Emotion	-0.081	[-0.150, -0.012]
Negative Emotion	0.035	[-0.017, 0.087]

13) Again, I thought that the authors’ handling of missing data was a strength. However, I would have liked to see descriptive data on the extent of missing data (range of % missing for variables in the analysis; number and percentage of complete cases).

Response: We appreciate the opportunity to clarify this. To be clear, we used multiple imputation to handle ‘item non-response’, which refers to cases when participants did not give a response to a survey question (Rubin, 2004, *Multiple Imputation for Nonresponse in Surveys*). We have now added Table S14 to illustrate the extent of item non-response. The percentage of complete data across all measures is 95.1% (i.e., 4.9% missing data), with a range of 75.0% (‘well-rested’ in the second survey of 2013) to ~100% (‘sex’, ‘age’, ‘health problems’). Supplementary Table 15 is added to the supplementary materials to present the percentage of complete responses by surveys.

We added the following to the method section:

“Overall, the percentage of complete data across all measures is 95.1% (i.e., 4.9% missing data; Supplementary Table 15 presents the missing data pattern).” (p. 27)

We would like to clarify that in 2009, 2010, 2011, and 2012, Gallup World Poll administered 2 surveys in Syria in the same calendar year, and each survey can include different items. For example, the item on the affordability of shelter was included in the first survey of 2010 but not in the second survey in 2010. When an item was excluded from a particular wave, we consider such missing data as ‘structurally missing’ (rather than item non-response), and did not carry out multiple imputation to account for structurally missing data. This is consistent with, for example, how structurally missing data were treated in a US Census Bureau dataset (Benedetto, Stinson, & Abowd, 2013).

Rubin DB. Multiple imputation for nonresponse in surveys. John Wiley & Sons; 2004 Jun 9.

Benedetto, G., Stinson, M. H., and Abowd, J. M. (2013). The creation and use of the SIPP Synthetic Beta.
https://www.census.gov/content/dam/Census/programssurveys/sipp/methodology/SSBdescribe_nontechnical.pdf.

14) For the cross-national analyses, I again wondered about the years of data

included. Was it again only 2008 and 2015? I assumed that this was to be consistent with the Syria-only analysis, but some justification would be useful, given that data were available from 2006 to 2016.

Response: Thank you for the question. We used all data available to provide the best estimates of cross-national trends in well-being. In Syria, all available data between 2008 and 2015 were used. In the other 162 countries, all available data from 2006 to 2016 were used. It is possible that our description of the coding of the linear time trend causes this confusion. To clarify, we coded the time trend using the following scheme: -0.29 = 2006; -0.14 = 2007; 0 = 2008; 0.14 = 2009;...; 1 = 2015; 1.14 = 2016.

The rationale is that the intercepts of the regression models represent well-being at 2008 (the beginning of the Syrian data collection in Gallup), and the coefficient for time can be interpreted as the total change in well-being from 2008 to 2015. The reviewer is correct that this is done to be consistent with the Syria-only analyses. We incorporated the reviewer's feedback and revised the relevant method section as follows:

"Using national-level data from Syria and 162 other countries, multilevel analyses were used to predict each well-being measure from survey year (-0.29 = 2006; -0.14 = 2007; 0 = 2008; 0.14 = 2009;...; 1 = 2015; 1.14 = 2016), dummy-coded variables for WHO regions (Syria as the reference group), and the interaction between survey year and regions. Random intercepts and random slopes for survey year at the country level were included. The rationale of the coding scheme for time is that the intercepts of the regression models represent well-being at 2008 (the beginning of the Syrian data collection in Gallup), and the coefficient for time can be interpreted as the total change in well-being from 2008 to 2015." (p. 27-28)

Results

15) Overall, I found the Results to be clearly written. One aspect that was confusing was that the analysis of the social support interactions seemed to be written as if there were longitudinal participant data ("the decline in well-restedness was sharper for participants with social support..."). This should be revised to be consistent with a serial cross-sectional design.

Response: We have revised the results section accordingly (p.11-12). "For example, Syrian participants with social support were 46% (OR=1.46, 95% CI 1.22 to 1.75) more likely to be well-rested prior to the conflict, but this association weakened during the conflict (ratio of ORs=0.50, 95% CI 0.15 to 0.85; relative excess risk due to interaction=-0.54, 95% CI -0.90 to -0.18)."

16) Did the results of the complete cases sensitivity analyses differ at all from the analyses with multiple imputation?

Response: The results are available in the supplementary materials. Supplementary Table 6 is based on results using multiple imputation, which can be compared to Supplementary Table 7, which is based on complete data. Similarly, Supplementary Table 9 can be compared to Supplementary Table 10. The analyses presented in Supplementary Tables 6 & 7 concern whether direct exposure to the Syrian conflict worsened well-being, and the key variable of interest is “Exposure”. Supplementary Tables 9 and 10 concern how social support may buffer the impact of the conflict, and the key variable is the ‘Year X Support’ interaction. Out of 25 pairs of significance tests (13 pairs in Supplementary Tables 6 and 7 and 12 pairs in Supplementary Tables 9 and 10), we note that 23 pairs showed consistent results in terms of statistical significance. In the 2 cases where the statistical significance differed (‘Hope’ and ‘Respect’ for Supplementary Tables 6 and 7), the point estimates are highly similar. For example, based on the analyses with multiple imputation (Supplementary Table 6), the regression coefficient for ‘exposure’ in predicting ‘hope’ is -0.327 (95% CI [-0.597; 0.057]), and the regression coefficient based on complete case analyses (Supplementary Table 7) is -0.333 (95% CI [-0.597; -0.069]). The statistical significance changed because the 95% CIs are wider in analyses based on multiple imputation (to account for uncertainty due to missing data). Overall, we interpret the results as highly consistent. We added the following to the results section: “Sensitivity analyses based on complete-case analyses showed very similar results (Tables S7 and S10).”

As an additional note, we apologise for making a copyediting error in the original supplementary materials, where the results for the ‘Respect’ and ‘Freedom’ columns for Supplementary Tables 7 and 10 were flipped. We have corrected this error in the revised supplementary materials.

Discussion

17) Can the authors define “societal spillover effects” and provide a citation for this phenomenon?

Response: We have rephrased the sentence to clarify the meaning of societal spillover effect. “Our finding suggests societal spillover effects, wherein well-being declined to a similar degree for participants with and without direct exposure to the conflict.” (p. 18)

In other disasters (e.g., 9/11), researchers have also documented that the impact of disasters can go beyond individuals who were directly exposed to the event. We also added a reference to the following reviews:

Galea S, Nandi A, Vlahov D. The epidemiology of post-traumatic stress disorder after disasters. *Epidemiologic reviews*. 2005 Jul 1;27(1):78-91.

18) Can the authors define and provide a citation for “wartime contextual effect”?

Response: Both Reviewers 2 & 3 commented on the wartime contextual effect. Reviewer 3 noted that this appears to be a case of response shift/recalibration. This prompted us to review again the literature on judgement and decision making. A recent paper published in *Science* (Levari et al., 2018) demonstrated ‘prevalence-induced concept change’ as a feature in human judgement. The basic premise is that how humans interpret the presence or absence of an object or a concept depends on its prevalence. For example, the authors showed that in a visual perception task, as the prevalence of blue dots on a screen become rarer, individuals would start categorising purple dots as blue.

The prevalence-induced concept change can be used to understand the counter-intuitive findings on health problems. For instance, the widespread violence in Syria led to a lower prevalence of healthy individuals. In turn, the lowered prevalence may lead participants to start categorizing ‘individuals with minor health problems’ as ‘healthy individuals’. We have rephrased the discussion section to address this.

“Second, human judgment is context dependent, and recent research has shown ‘prevalence-induced concept change’ as a characteristic of human judgement. When applied to the context of the Syrian conflict, prevalence-induced concept change can refer to the tendency to lowering the threshold for ‘health’ when fully healthy people become less prevalent. In other words, participants may not report relatively minor pain or health problems when people around them have lost their lives, experienced chemical attacks, or suffered severe injuries.” (p. 19)

Levari DE, Gilbert DT, Wilson TD, Sievers B, Amodio DM, Wheatley T. Prevalence-induced concept change in human judgment. *Science*. 2018 Jun 29;360(6396):1465-7.

19) The authors might consider including the use of single items, mostly dichotomous, among the limitations. Although these did facilitate cross-cultural comparisons, future research should draw on more in-depth measures with stronger psychometric properties, particularly for the within-country analysis.

Response: We agree with the reviewer and added this as a limitation. “Second, the physical, mental, and social well-being indicators analysed here were limited by data availability in the Gallup World Poll, and the use of mostly single-item and dichotomous self-reported outcome measures is a limitation compared to validated gold-standard multi-item instruments of well-being. However, use of objective records to assess well-being was

impeded by the wide-spread destruction of health care facilities, and indeed, the United Nations discontinued reporting the death tolls in the Syrian conflict, citing the lack of reliable sources.^{6,9,10} Therefore, the current use of self-reported well-being measures covering topics related to health care, physical health, mental well-being, and social support complement past studies and provide useful, albeit imperfect, insights into life during the Syrian conflict. Moreover, Gallup's use of a reasonably brief survey with similar items being asked repeatedly in many countries allows for cross-temporal and cross-national comparisons. Future research should draw on more sophisticated measures to track Syrians' well-being over time." (p. 20)

Supplemental

20) Table S4 – Were ORs excluded for dependent variables for which the omnibus test was non-significant? I would recommend clarifying in the table note. Also, can confidence intervals be added to this table?

Response: Thank you for the suggestion. We have added the confidence intervals to the table. After adding the confidence intervals, the table is too large, and we split it into Supplementary Tables 3 and 4.

We have additionally revised the results within the table. FC (the first author) repeated the analyses but could not reproduce the exact results in the original Table S4. The Supplementary Tables 3 and 4 showed the updated results, and the manuscript texts have also been updated accordingly.

"From 2008 to 2015, participants from Rural Damascus (where the death toll was highest based on estimates from 2011 to 2014) were 15 times (95% CI 13.9 to 17.5) less likely to report satisfaction with access to healthcare, compared to 3.6 times (95% CI 3.1 to 4.2) for the overall country. Aleppo, where the battle accounted for about one-tenth of the death toll and caused most damage to hospitals, showed the sharpest decline in hope from 2008 to 2015, and the extent of the estimated decline (-2.22 unit, 95% CI -2.29 to -2.16) was about 10 times that of Ar-Raggah (-0.20 unit, 95% CI -0.55 to 0.16)." (p. 11)

21) Table S11 – I would recommend also listing in bold when Syria was among the *top* five countries on negative indicators (e.g., health problems, physical pain).

Response: We incorporated the reviewer's suggestion in Supplementary Table S11.

Reviewer #3 (Remarks to the Author):

Title Manuscript: Physical, mental, and social well-being in Syria and 162 countries from 2006 to 2016

Overall

This is a very important and well-written manuscript, that describes the social and health effects of the Syrian war on the Syrian population in relation to other conflict and disaster-affected areas in the world. The authors used the data of the Gallup World Poll, a poll that includes Syrian samples (with Ns of 1002 to 2100 varying between waves) in which data such as physical protection/ shelter, access to health care and well-being were measured at several intervals between 2008-2015. To this reviewer's knowledge, these data have not been used for this purpose before, and they provide a very important insight in the situation of the people within Syria in these extremely adverse times.

Response: Thank you very much for your positive feedback.

Abstract

It would be correct to describe in the abstract that the Gallup World Poll data were used, in order to facilitate researchers when systematically searching for studies using this dataset.

Response: We have added Gallup World Poll to the abstract.

It is stated that "Syrians regardless of the degree of exposure were similarly affected". This is a rather bold statement since only a few exposures were measured. No data are available on sexual violence (information which Syrians may not easily disclose anyway due to stigma), and on more subtle forms of exposure to trauma with high impact such as witnessing atrocities.

Response: We agree with the reviewer's assessment, and there are many kinds of direct exposure that could heavily impact Syrians' well-being. We have toned down the sentence to emphasize the self-reported nature of the measures. "Syrians who reported being exposed to the conflict were similarly affected compared to those without direct exposure, suggesting country-wide spillover effects."

Introduction

The introduction is adequate, but would certainly benefit from a more grounded presentation of the variables studied. Why were variables such as social support selected, or access to health care, and for instance economic or financial circumstances were not? A more balanced overview of the vast literature on health and well-being in response to war and crisis should be presented, that should have guided the authors in deciding which variables to select.

Response: Thank you for the suggestion to present a more comprehensive review of the literature on health and crisis. The goal of our study is to evaluate the health impact of the Syrian conflict, and our focus on physical, mental, and social well-being was motivated by the World Health Organization's holistic definition of health. Together with Reviewer 2's comments about the introduction, we have completely reframed the introduction and included references to the literature on war and crisis.

"The Syrian conflict has widespread consequences on civilian health and well-being beyond mortality and displacement. Due to widespread destruction of health care facilities, progress in infant mortality has reversed, infectious diseases have been on the rise, and patients with chronic diseases have lost access to treatment. The damage to residential buildings and food distribution systems disrupt the basic needs for shelter and food. Exposure to war-related brutality is a salient risk factor for debilitating psychological conditions, with long-term impact that usually persists beyond the course of the war. Armed conflicts also destroy families and communities and force civilians into displacement, thus severely compromising social capital. Based on prior findings from major disasters, younger age, being female, exposure to the conflict, and lack of social support are additional risk factors for well-being decrement. However, objective assessment of population well-being is obstructed by the unprecedented "weaponisation of health care" (i.e. the denial of access to health care as a strategy of war). Therefore, Syrians' self-reported well-being could be particularly relevant to measuring an important aspect of the impact of the conflict. Based on the World Health Organization's holistic definition of health, we sought to assess the changes in self-reported physical, mental, and social well-being and to characterize well-being by sex, age, social support, and exposure to the conflict.

Although a large corpus of research has focused on individual major population events, comparison across different events is rarely conducted. Notably, in mental well-being research, studies have found little to no long-term decrease following the 2007-2008 global financial crisis, 2011 Fukushima disaster, or 2015 Paris terrorist attack. These findings give evidentiary support to an influential hypothesis in the psychology literature that life circumstances may play a limited role in changing long-term well-being. However, these findings do not appear to be consistent with empirical research on disaster and health. To clarify such apparent divergence, we adopted a global perspective to evaluate the changes in well-being in Syria relative to the rest of the world, including comparisons with other countries' experiences with armed conflict, social unrest, and natural disasters." (p.3-4)

Results

Table 1 shows that some variables do not decline as dramatically as one would

expect within these circumstances. An example is hope, which drops only 1 point on a scale of 0 to 10 from pre-conflict to during the conflict.

Response: We understand the reviewer's comment regarding the magnitude of the changes in well-being. Researchers in different areas may have different expectations of how much well-being should drop in these contexts. We provided the global comparisons as an additional way to place the changes in well-being in Syria into context. Using the reviewer's example, we note that a drop of 1.3 unit in hope is already the sharpest drop observed in all countries surveyed by the 2006-2016 Gallup World Poll. Indeed, for 11 out of 13 outcome variables, the drops in Syria were the largest declines observed in the dataset (Table 2). Accordingly, our interpretation is that these declines in well-being are substantial. We revised part of the first paragraph of the discussion to highlight this (p. 17):

“Consistent with reports of rising mortality, blatant violations of international laws, and compromised safety for healthcare workers, the current study found that a wide spectrum of physical, mental, and social well-being has dropped in Syria during the conflict. However, the extent of the declines can be difficult to interpret without comparison (e.g., how large is a drop of 1.3 unit in hope?). We therefore provided the cross-national analyses to place the changes in well-being in Syria into global context. Notably, prior to the conflict, Syria's level of well-being was comparable to neighboring countries. However, from 2006 to 2016, the substantial and pervasive declines in well-being in Syria were unmatched when compared to the eastern Mediterranean and other WHO regions. Changes in well-being in Syria are therefore not explained by the broader regional or global trends. Indeed, the fall in most well-being indicators in Syria has exceeded all countries in the Gallup World Poll, even in comparison to countries that have also suffered military conflict, civil strife, and natural disasters, including Haiti, Sudan, and Iran.”

Interestingly, “health problems compared to others” already dropped before the start of the conflict, in 2010, and it did not increase back to the pre-conflict level in the period from 2011-2015. How would the others explain this? Did other changes occur in the Syrian society that could explain this result?

Response: We understand the reviewer's observation on health problems, where it dropped in 2010 and remained relatively stable from 2011 to 2015. It is indeed puzzling. Our working hypothesis of a 'war time contextual effect' (now revised as 'prevalence-induce concept change' to be more consistent with recent research in human judgment) is described in the discussion section. A recent paper published in *Science* (Levari et al., 2018) demonstrated 'prevalence-induced concept change' as a feature in human judgement. The basic premise is that how humans interpret the presence or absence of an object or a concept depends on its prevalence. For example, the authors showed that in a visual

perception task, as the prevalence of blue dots on a screen become rarer, individuals would start categorising purple dots as blue.

The prevalence-induced concept change can be used to understand the counter-intuitive findings on health problems. For instance, the widespread violence in Syria led to a lower prevalence of healthy individuals. In turn, the lowered prevalence may lead participants to start categorizing 'individuals with minor health problems' as 'healthy individuals'. We have rephrased the discussion section to address this.

“Second, human judgment is context dependent, and recent research has shown ‘prevalence-induced concept change’ as a characteristic of human judgement. When applied to the context of the Syrian conflict, prevalence-induced concept change can refer to the tendency to lowering the threshold for ‘health’ when fully healthy people become less prevalent. In other words, participants may not report relatively minor pain or health problems when people around them have lost their lives, experienced chemical attacks, or suffered severe injuries.” (p. 19)

Levari DE, Gilbert DT, Wilson TD, Sievers B, Amodio DM, Wheatley T. Prevalence-induced concept change in human judgment. *Science*. 2018 Jun 29;360(6396):1465-7.

The reviewer also noted that the drop began in 2010 (prior to the conflict). It is perhaps important to note that in 2007-2010, Syria experienced a drought, which contributed to economic downturn and a reduced supply for food and water. These precursors to the full-blown conflict could have altered the context through which participants reported their health problems. Having said that, it is difficult to fully explain the observed trend with the data that we have. Overall, the goal of our study is to understand the total change in physical, mental, and social well-being from before the Syrian conflict to during it. Although we acknowledge the trends in health problems and physical pain are counterintuitive, our findings did demonstrate large declines in a wide spectrum of physical, mental, and social well-being outcomes.

A comparison has been done with 162 other countries. In the methods (under “Classification of countries”) it is stated that there was a focus on comparisons between Syria and other countries in the Eastern Mediterranean, but this is not reflected in the results.

Response: Thank you for the opportunity to clarify this point. On p.9 of the manuscript, a sub-section of the results section compared Syria’s well-being to the rest of the Eastern Mediterranean region. Based on the reviewer’s feedback, we now recognize that this comparison may not have been the ‘focus’ (as we also present comparisons with all other 162 countries and comparisons with other countries in turmoil). We have rephrased the method section to soften the tone. “We compared well-being changes

between Syria and the eastern Mediterranean region, as geopolitically they are more similar than other WHO regions.⁶” (p. 25)

Discussion

The crisis in Syria started in 2011. From 2015, the Syrian conflict became an international conflict, involving other countries including Turkey and Russia. This is also the year in which the refugee streams to Europe were at its peak. This was also the year in which the last measurement wave was carried out. It would be interesting if the authors would reflect more in-depth on how the timing of the assessments in relation to the political situation on the ground may have affected the results. In addition, the situation has further evolved after 2015, do the authors expect the same outcomes if they would have included more recent data?

Response: We agree with the reviewer that it is important to consider the broader context. The reviewer is correct that the Syrian conflict has further escalated in international scope. Therefore, Syrians’ physical, mental, and social well-being likely remain near (if not worsen past) post-conflict levels as observed in our study. However, we also cannot rule out the possibility of localised recovery (e.g., in Raqqa, where ISIS force was driven out). We incorporated the reviewer’s comment and added a discussion of this possibility in the discussion section.

“Importantly, 2015 was the last data collection in Syria by the Gallup World Poll. Since then, the Syria conflict has grown in complexity, marked by escalation in international involvement but also the recapture of the city of Raqqa from ISIS’ control. Future research should continue to track Syrians’ well-being through these major developments.” (p. 18)

Another issue that would deserve further elaboration and discussion, is whether the results differ between regions in Syria. Regions that may have been under ISIS control may be differently affected than regions under the control of the Syrian government. This might have had a direct effect on important variables such as access to health care.

Response: Results by regions (governorates) are presented in Figure 1 and Supplementary Tables 3 and 4. We observed significant heterogeneity in changes of well-being by governorates. We have considered the possibility of examining areas under ISIS control. However, we rather suspect that it would be exceedingly difficult to collect high-quality data from such geographical regions given the situation (although Gallup did not specifically mention this as an exclusion criterion). In addition, the development of the Syrian conflict is complex, and a single governorate can include territory occupied by different groups. Therefore, a proper study on the impact of ISIS would require a much

more detailed measurement of participants' location than our current measure (at the governorate-level).

The surprising finding that some variable improved instead of worsened, has been explained by the authors by a sampling bias and “war time contextual bias” (the latter being a form of a response shift or recalibration). However, the sampling bias would have affected all variables measured, and as such, this explanation undermines all other effects found in this study, also the intuitive ones. Thus, the authors should do a better job in explaining these specific improvements, or lack of clear deterioration. As stated earlier, are other factors present that may explain why self-reported health problems are not increasing? And perhaps reported pain was not decreasing since Syrians may have still had still access to pain medication, despite the reduced access to health care. It is commonly known that Syrians, even when displaced to other countries, still have contact with Syrian doctors, who may prescribe medications. It would be worthwhile to include a Syrian professional in the author team, who might provide a more informed and in-depth understanding of the specific effects found. Or the results could have been presented to a focus group of Syrian (refugee) stakeholders, who might provide further clues to the interpretation. Such a mixed-methods approach would certainly benefit the paper.

Response: We thank the reviewer for the comment, and we agree that the sampling bias and ‘war time contextual bias’ could have affected all variables measured. In this regard, we have added the following to the discussion section to acknowledge this potential bias on all outcomes: “Moreover, we acknowledge that the sampling bias and prevalence-induced concept change, however unlikely, could have also biased the other 11 outcomes.” (p. 19)

We also agree with the reviewer with regards to 1) the importance of understanding specifically why self-reported health problems and pain did not deteriorate and 2) the value of a mixed-method approach. However, the Gallup World Poll did not ask questions on pain medication or healthcare usage. For a mixed-methods approach, such an exercise should be a separate full paper to tease out the qualitative elements of how the Syrian conflict impacted physical, mental, and social well-being. We believe this is out of the scope of our current manuscript, which aims to examine the total changes in well-being before and during the Syrian Conflict. We point to these as future research directions: “Future research should consider verifying and explaining these counter-intuitive results as underreporting of health issues could delay proper medical treatment.” (p. 19)

A further issue to discuss is whether cultural differences between samples might have affected the results, specifically the between-country comparisons. Are the authors confident that variables such as health problems and well-being can be

reliably and validly measured across cultural settings with the same questions? This might be a potential limitation, especially when comparing countries.

Response: Thank you for pointing out this potential limitation. Cultural differences indeed could have made cross-national comparisons difficult. We added this as a limitation:

“Third, although Gallup’s use of a reasonably brief survey facilitates cross-temporal and cross-national comparisons, the potentially substantial cultural differences across the globe could have affected the results.” (p. 20)

However, there is some evidence for the cross-cultural validity of the measures used in the Gallup World Poll. For example, in Chapter 2 of the 2018 World Happiness Report (commissioned by the UN), international migrants are found to have more similar levels of well-being as native-borns in the host countries than residents in their original countries. If self-reported well-being is based largely on cultural differences, we would expect that international migrants would report well-being at similar levels as their original country. Therefore, self-reported well-being appears to be responsive to the changing circumstances (in addition to cultural values).

Access to health care is a very salient variable in this study, dropping dramatically from 2011 onwards. The authors conclude that along with providing accelerating the peace process, psychosocial interventions should be offered to Syrians affected. It will be challenging to provide psychosocial interventions to Syrians when the war is ongoing, but efforts are made to train non-specialist helpers in delivering mental intervention in low resource areas where access to health care is extremely limited. An example is the scaling-up of the evidence-based WHO PM+ programme across Syrians in Europe and the Middle East (e.g., Sijbrandij et al., Strengthening mental health care systems for Syrian refugees in Europe and the Middle East: integrating scalable psychological interventions in eight countries, Eur J Psychotraumatology, 2017, 8 (2))

Response: We agree with the reviewer and added the suggested reference into the discussion. “The large psychological impact of the Syrian conflict reinforces the importance of scaling up evidence-based psychological interventions (e.g., Problem Management Plus developed by the WHO) in Syria and neighbouring countries with sizeable populations of Syrian refugees.” (p. 18)

Methods

The Gallup dataset may be more extensively described (how it is collected, for what purpose, by whom, etc.)

Response: Thank you for the suggestion. We elaborated on the Gallup World Poll in the beginning of the revised method section.

“This study was a secondary data analysis of de-identified data from the Gallup World Poll, which adopted a serial cross-sectional design, with multiple waves of data collection in Syria from 2008 to 2015 and 162 other countries from 2006 to 2016. The goal of the Gallup World Poll is to collect reliable data on attitudes, opinions, and well-being of randomly selected, population-representative samples across the world. The inclusion criteria of the Gallup World Poll are non-institutionalised civilians aged 15 or above. Annually, the Gallup World Poll surveys about 1,000 participants in each society (Mean=1,187, SD=844, range=500 to 13,408) with a total sample of 1,722,558 participants across 163 territories (including Syria).

Random-digit-dialing telephone interviews were conducted in high-income countries (defined as where landline phones can reach at least 80% of the population).¹³ Face-to-face interviews were conducted in low-income countries, including Syria.¹³ The samples were obtained using stratified random sampling, where primary sampling units (PSU) were sampled in subnational units (governorate in the case of Syria) such that sample sizes were proportionate to the populations of the subnational units.¹³ After a PSU was sampled, a random-route method was used to select households, with at least 3 interview attempts.¹³ Household members were randomly sampled using Kish grid.¹³ Interviewers were provided training on administering the questionnaire and on executing the randomisation procedure with quality control from supervisors.¹³ (p. 21-22)

The authors state that since the data were secondary data, no ethics approval was obtained from accredited ethics review boards in St Louis or Hong Kong. Nevertheless, the authors should provide a formal waiver of at least one of these committees.

Response: We share the reviewer’s commitment towards ethical research. According to Title 45 in the Code of Federal Regulations Part 46 set forth by the U.S. Department of Health and Human Services, the current study did not constitute human subject research because the data are de-identified secondary data, and the current researchers did not interact with any respondents.

According to the ‘Investigator’s code of practice’ of the Institutional Review Board of the University of Hong Kong/ Hospital Authority Hong Kong West Cluster, the scope of IRB is over ‘research on materials of human origins, such as body tissue and fluid, including “waste” or “leftover” from diagnosis, treatment and post-mortem examination, or archiving such materials for future studies, and collation of records/data (whether prospective or retrospective) where there is a reasonable likelihood that such may link to the individuals’

identifiable particulars or identifiers'. Therefore, analysing de-identified data does not fall into the scope of the IRB.

More information should be provided on how physical wellbeing was measured. Which the questions were asked, and on which scale were the responses scored?

Response: Given that two reviewers suggested that we provide additional details about the measures, we have decided to move the original Table S1 into the main text as Table 1 (p. 5-6). A truncated version of it is reproduced here:

Items	Response options
Physical Well-being	
Health Problem	
Do you have any health problems that prevent you from doing any of the things people your age normally can do?	Yes/No
Physical Pain	
Did you experience the following feelings during a lot of the day yesterday? How about physical pain?	Yes/No
Well-restedness	
Did you feel well-rested yesterday?	Yes/No
Health Care	
In the city or area where you live, are you satisfied or dissatisfied with the availability of quality healthcare?	Satisfied/Dissatisfied
Food	
Have there been times in the past 12 months when you did not have enough money to buy food that you or your family needed?	Yes/No
Shelter	
Have there been times in the past 12 months when you did not have enough money to provide adequate shelter or housing for you and your family?	Yes/No

****REVIEWERS' COMMENTS:**

Reviewer #1 (Remarks to the Author):

I served as a reviewer on the original submission, and my evaluation was that the paper was excellent--I had only minor suggestions that I felt the authors could take or leave. To avoid presenting authors with a moving target, I always try to accept review requests of revised manuscripts, while confining myself to the issues I originally raised. In this case I'm not sure it was really necessary for me to review the paper again, given the nature of my original comments. That said, I believe the authors responded to my comments in a perfectly satisfactory way, and I hope to see this important paper appear in this journal.

Signed,
Elizabeth Dunn

Reviewer #2 (Remarks to the Author):

Thank you for the opportunity to review the revised version of the manuscript "Physical, Mental and Social Well-Being in Syria and 162 Other Countries from 2006 to 2016" for Nature Communications. I continue to think that this is a very strong and meaningful manuscript. The authors have further done an impressive job thoroughly addressing the comments of the three reviewers. I found the manuscript to be much clearer and compelling as a result of the changes the authors made. Bravo!

My only remaining comment (and it is a minor one) pertains to the Introduction section. I greatly appreciated the efforts the authors made to expand upon their introduction section. However, I still would have liked to see a bit more text to provide a rationale for some of the (very compelling) analyses the authors did. For example, I found it fascinating that the association between social support and wellbeing declined during the conflict. I wondered if there was any precedent for exploring whether its impact varied over the course of time, and in particular whether any prior research has shown that associations dissipate during times under which an entire community is under duress.

Reviewer #3 (Remarks to the Author):

Many thanks for the revised manuscript.

The authors sufficiently addressed the issues of the reviewers, and this version has certainly improved.

There are still two outstanding issues for me as a reviewer:

-I think the answer to the issue that a Syrian expert group should have been involved, has not

appropriately been addressed. Of course I understand that this would be time investment that the author would not like to take. It would definitely not be a study or paper on itself as the author suggest. Within global mental health research, mixed methods approaches are the golden standard.

-I am still not convinced that this study was exempted from obtaining a waiver for ethics approval. However, I leave it up to Nature to decide whether adequate ethics procedures were followed in order to get approval for analysing these data.

Point-by-point response

Reviewers' comments:

Reviewer #1 (Remarks to the Author):

I served as a reviewer on the original submission, and my evaluation was that the paper was excellent--I had only minor suggestions that I felt the authors could take or leave. To avoid presenting authors with a moving target, I always try to accept review requests of revised manuscripts, while confining myself to the issues I originally raised. In this case I'm not sure it was really necessary for me to review the paper again, given the nature of my original comments. That said, I believe the authors responded to my comments in a perfectly satisfactory way, and I hope to see this important paper appear in this journal.

**Signed,
Elizabeth Dunn**

Response: Thank you again for your feedback on our manuscript!

Reviewer #2 (Remarks to the Author):

Thank you for the opportunity to review the revised version of the manuscript "Physical, Mental and Social Well-Being in Syria and 162 Other Countries from 2006 to 2016" for Nature Communications. I continue to think that this is a very strong and meaningful manuscript. The authors have further done an impressive job thoroughly addressing the comments of the three reviewers. I found the manuscript to be much clearer and compelling as a result of the changes the authors made. Bravo!

My only remaining comment (and it is a minor one) pertains to the Introduction section. I greatly appreciated the efforts the authors made to expand upon their introduction section. However, I still would have liked to see a bit more text to provide a rationale for some of the (very compelling) analyses the authors did. For example, I found it fascinating that the association between social support and wellbeing declined during the conflict. I wondered if there was any precedent for exploring whether its impact varied over the course of time, and in particular whether any prior research has shown that associations dissipate during times under which an entire community is under duress.

Response: Thank you for your constructive feedback on our manuscript! We share the reviewer's fascination about the findings on social support.

When we conducted the literature review, we found that in disasters, social support is consistently linked to fewer post-disaster psychological symptoms (e.g., Goldman & Galea, 2014; Neria, Nandi, & Galea, 2008). This is the a priori basis of our analysis of social support, as already cited in the Introduction. We are somewhat hesitant to now edit the Introduction based on what we found – a methodological fallacy some have characterized as ‘hypothesizing after the results are known’ (Kerr, 1998).

We note, however, that there is precedent (e.g, Cook & Bickman, 1990) documenting the lack of an association between social support and post-disaster mental health. We added this reference to the discussion (p.10).

Cook, J. D., & Bickman, L. (1990). Social support and psychological symptomatology following a natural disaster. *Journal of Traumatic Stress, 3*(4), 541–556.

Goldmann, E., & Galea, S. (2014). Mental health consequences of disasters. *Annual review of public health, 35*, 169-183.

Kerr, N. L. (1998). HARKing: Hypothesizing after the results are known. *Personality and Social Psychology Review, 2*(3), 196-217.

Neria, Y., Nandi, A., & Galea, S. (2008). Post-traumatic stress disorder following disasters: a systematic review. *Psychological medicine, 38*(4), 467-480.

Reviewer #3 (Remarks to the Author):

Many thanks for the revised manuscript.

The authors sufficiently addressed the issues of the reviewers, and this version has certainly improved.

There are still two outstanding issues for me as a reviewer:

-I think the answer to the issue that a Syrian expert group should have been involved, has not appropriately been addressed. Of course I understand that this would be time investment that the author would not like to take. It would definitely not be a study or paper on itself as the author suggest. Within global mental health research, mixed methods approaches are the golden standard.

Response: We agree with the reviewer regarding the importance of a mixed-method approach. We are also grateful for the reviewer’s understanding of the resources involved in such an endeavor. In the current manuscript, we aim to document the total changes in physical, mental, and social well-being through the Syrian Conflict, and we believe our quantitative overview has provided valuable albeit preliminary insight to this overall aim.

We remain hopeful that the current manuscript will bring additional attention to the Syrian Conflict and motivate future substantive enquiries using both qualitative and quantitative methods.

-I am still not convinced that this study was exempted from obtaining a waiver for ethics approval. However, I leave it up to Nature to decide whether adequate ethics procedures were followed in order to get approval for analysing these data.

Response: We share the reviewer's commitment towards research ethics, and we appreciate the opportunity to clarify the issue. The analyses took place in the US and in Hong Kong. According to Title 45 in the Code of Federal Regulations Part 46.102(e), the definition of *human subject* is as follows:

(e)(1) Human subject means a living individual about whom an investigator (whether professional or student) conducting research:

*(i) Obtains information or biospecimens through **intervention or interaction** with the individual, and uses, studies, or analyzes the information or biospecimens; or*

*(ii) Obtains, uses, studies, analyzes, or generates **identifiable** private information or **identifiable** biospecimens.*

The Gallup World Poll data were collected by Gallup and its partners. The Gallup World Poll data that we received from Gallup are **de-identified**. The current investigators **did not interact with or intervene on** the participants in the Gallup World Poll. Therefore, the secondary data analyses reported in our manuscript do not constitute human subject research.

In the previous round of response, we provided the scope of the Institutional Review Board of the University of Hong Kong/ Hospital Authority Hong Kong West Cluster. It covers 'research on materials of human origins, such as body tissue and fluid, including "waste" or "leftover" from diagnosis, treatment and post-mortem examination, or archiving such materials for future studies, and collation of records/data (whether prospective or retrospective) **where there is a reasonable likelihood that such may link to the individuals' identifiable particulars or identifiers**'. Thus, analyzing de-identified data does not fall into the scope of the IRB.